# Grounding or Guessing? Visual Signals for Detecting Hallucinations in Sign Language Translation

**Yasser Hamidullah**[1,2]    **Koel Dutta Chowdhury**[2]    **Yusser Al Ghussin**[1,2]    **Shakib Yazdani**[1]
**Cennet Oguz**[1,2]    **Josef van Genabith**[1,2]    **Cristina España-Bonet**[1,3]

[1]German Research Center for Artificial Intelligence (DFKI GmbH),
[2]Saarland University, Saarland Informatics Campus, Germany
[3]Barcelona Supercomputing Center (BSC-CNS), Barcelona, Catalonia, Spain

## Abstract

Hallucination, where models generate fluent text unsupported by visual evidence, remains a major flaw in vision–language models and is particularly critical in sign language translation (SLT). In SLT, meaning depends on precise grounding in video, and gloss-free models are especially vulnerable because they map continuous signer movements directly into natural language without intermediate gloss supervision that serves as alignment. We argue that hallucinations arise when models rely on language priors rather than visual input. To capture this, we propose a token-level **reliability** measure that quantifies how much the decoder uses visual information. Our method combines feature-based sensitivity, which measures internal changes when video is masked, with counterfactual signals, which capture probability differences between clean and altered video inputs. These signals are aggregated into a sentence-level reliability score, providing a compact and interpretable measure of visual grounding. We evaluate the proposed measure on two SLT benchmarks (PHOENIX-2014T and CSL-Daily) with both gloss-based and gloss-free models. Our results show that reliability predicts hallucination rates, generalizes across datasets and architectures, and decreases under visual degradations. Beyond these quantitative trends, we also find that reliability distinguishes grounded tokens from guessed ones, allowing risk estimation without references; when combined with text-based signals (confidence, perplexity, or entropy), it further improves hallucination risk estimation. Qualitative analysis highlights why gloss-free models are more susceptible to hallucinations. Taken together, our findings establish reliability as a practical and reusable tool for diagnosing hallucinations in SLT, and lay the groundwork for more robust hallucination detection in multimodal generation.

## 1 Introduction

Despite being a video-to-text task, sign language translation (SLT) differs fundamentally from other video understanding problems. Whereas most multimodal approaches focus on recognizing scenes or describing visual events, SLT must process a complete natural language, namely sign language, with its own vocabulary and grammar. Early approaches relied on glosses, which are explicit human annotations that map video to textual sign labels (Camgoz et al., 2018; 2020). Glosses provide strong supervision but limit scalability, since most large-scale SLT datasets are only weakly labeled and lack gloss annotations.

Recent advances in SLT increasingly favor gloss-free architectures that rely on large language model backbones (Gong et al., 2024; Chen et al., 2024b; Hwang et al., 2025). While this shift alleviates the costly gloss annotation bottleneck, it also changes the nature of mistakes these systems make. We show that when visual representations are weak whether due to ambiguous signing or low video quality, the language model tends to dominate decoding. This results in translations that may appear fluent yet fail to reflect the signer's actual message, resulting in hallucinations. Such behavior mirrors findings in large vision–language models (LVLMs), where hallucinations often arise from the model's

tendency to prioritize language patterns over visual evidence (Ghosh et al., 2025; Parcalabescu & Frank, 2025).

In an ideal SLT system, strong visual representations ground the translation, while the language model complements them by ensuring grammaticality and fluency. Hallucinations emerge precisely when this balance is disrupted in favor of the language model.

Hallucination detection has been studied in other multimodal settings, particularly in LVLMs and image/video to text generation, where strong language priors can override weaker visual signals (Chen et al., 2025; Min et al., 2025). In these domains, approaches often focus on aligning model attention with visual content or using perturbation-based analyses to identify unreliable outputs. However, SLT presents a more challenging scenario because the visual modality is not auxiliary (more than just understanding a scene, detecting objects and actions) but constitutes the source language itself. Consequently, hallucinations in SLT are directly equivalent to translation errors, making the detection and quantification of visual grounding critical. In this work, we define **hallucination in SLT** as the generation of content tokens that are not supported by the signed video.

A crucial first step toward mitigating and reducing hallucinations is the ability to characterize and detect them, and to assign reliability scores that allow ranking candidate translations by their risk of hallucination. Text-based signals such as confidence, entropy, and perplexity capture useful linguistic uncertainty, but they miss a critical aspect in SLT: whether the output is actually supported by the signed input. We therefore introduce a complementary dimension of hallucination detection we refer to as **grounding usage** which quantifies the extent to which generated tokens rely on video evidence rather than language priors. By quantifying grounding usage and combining this signal with established text-only baseline measures, we can identify hallucinations arising when models guess from language model priors instead of leveraging the video stream. To this end, we define a **reliability score** that compares standard decoding with counterfactual runs where the video is masked or replaced. These perturbations yield two classes of evidence: (i) changes in internal states and routing (e.g., hidden-state angular shifts, cross-attention activations), and (ii) probability margins between clean and counterfactual outputs. A linear fusion of the two types produces token-level reliabilities, which are then pooled into sentence-level scores. Our experiments across datasets (PHOENIX-2014T (Camgoz et al., 2018) and CSL-Daily (Zhou et al., 2021)) and architectures (T5 (Hwang et al., 2025) and mBART (Chen et al., 2022)) demonstrate that reliability not only surpasses text-only baselines in predicting hallucinations but also generalizes robustly in cross-dataset and cross-architecture transfer. Specifically, it achieves an accuracy of 97% and maintains competitive regression performance when approximating the level of hallucination rate, with a Spearman correlation of $\rho = 0.72$ (between predicted and computed hallucination rates). Our findings further reveal that, compared to gloss-based models, gloss-free models hallucinate more often because they systematically under-utilize visual information, defaulting to linguistic priors rather than grounding in the video. Our contributions and findings can be summarized as follows:

- To the best of our knowledge, this is the first work to investigate hallucination in SLT.
- We introduce a novel measure referred to as reliability based on *grounding usage* that quantifies how much generated tokens rely on visual input, providing a direct measure of whether SLT outputs are grounded in the source video.
- We design a reliability scoring framework using counterfactual perturbations, capturing both internal model changes and output probability shifts.
- We empirically demonstrate that reliability surpasses text-only baseline measures in detecting hallucinations and show why gloss-free models are particularly prone to them.

## 2 RELATED WORK

Hallucinations are often characterized as the generation of content that is logically unsound or that departs from the source; in other words, model outputs that assert facts, events, or details not supported, or even contradicted, by the available evidence or ground truth (Ji et al., 2023). In multimodal learning, hallucinations have been widely studied in image and video captioning, particularly with respect to object mentions. For example, Rohrbach et al. (2018) introduced the CHAIR metric to detect and quantify hallucinated objects in MSCOCO image captions, showing that captioning models frequently insert visually ungrounded entities. Similar patterns have been

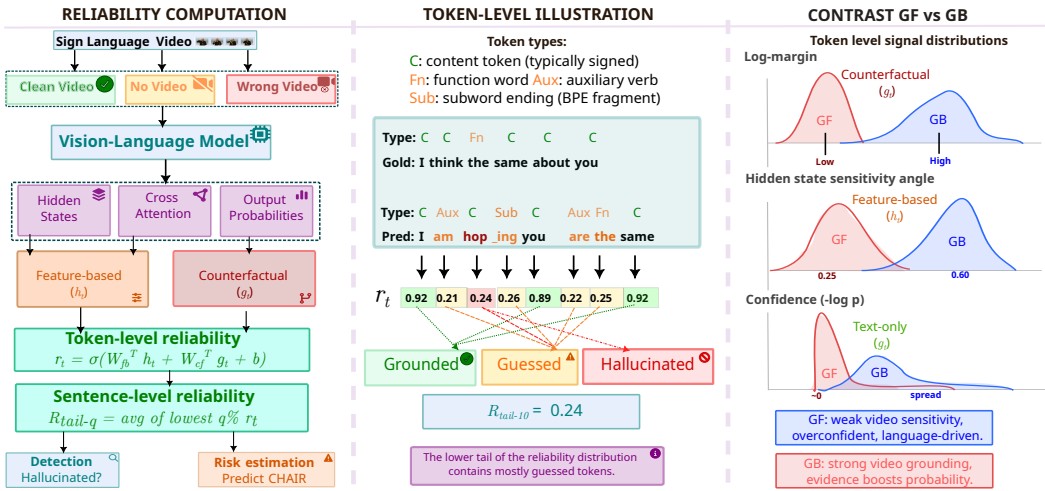

Figure 1: Overview of the reliability framework. **Left**: Token-level signals are extracted by comparing sensitivity under clean, no-video and wrong video inputs, quantifying each token's dependence on visual evidence. **Center**: Low reliability tokens correspond to weak visual reliance (guessed) — exactly where hallucination arises. **Right**: Signal distributions contrast the language-dominant behavior of gloss-free (GF) models with the visually grounded behavior of gloss-based (GB) models.

observed in video captioning, where generated descriptions may include unsupported objects, actions, or temporal details (Ullah & Mohanta, 2022; Liu & Wan, 2023).

With the advent of LVLMs, hallucination detection techniques have expanded significantly. Approaches now include reinforcement learning to align outputs with factual grounding (Gunjal et al., 2024), fact verification methods that leverage external knowledge bases or retrievers (Yin et al., 2024; Sahu et al., 2024), and tool-augmented strategies that integrate external APIs to assess factual consistency (Chen et al., 2024a). Another noteworthy case is SelfCheckGPT (Manakul et al., 2023), a sampling-based approach that detects hallucinations by leveraging response consistency: factual knowledge tends to yield stable outputs across multiple generations, whereas hallucinations produce divergent or contradictory answers.

## 3 FROM GROUNDING TO GUESSING: RELIABILITY AS A LENS ON VISUAL USAGE

**Intuition.** At each decoding step, a multimodal model can either *ground* the next token in the video or *guess* it from language priors or use a mixture of both. Our goal is a token-level reliability score $r_t$ that increases when the decision genuinely relies on the video signal. Prior work in hallucination detection has mostly relied on text-intrinsic proxies such as entropy or perplexity (Kuhn et al., 2023; Farquhar et al., 2024; Obeso et al., 2025), which are agnostic to visual input. While these capture uncertainty, they fail to tell whether a confident prediction is grounded in visual evidence or merely plausible under language priors. In multimodal generation tasks, where visual evidence is the *source* as in SLT, this distinction is critical (Chen et al., 2022; Guo et al., 2024). To capture visual grounding, we propose a **reliability** score. At each step, we (i) measure **feature-based sensitivity**, which captures how much the output hidden states and the cross-attention drift when the visual input is removed; and (ii) compare decoding with the correct input (which we call clean) to **counterfactual** runs in which the video is masked or mismatched. The core mechanism is illustrated in Figure 1.

Intuitively, feature-based signals reflect *internal reliance* on visual context, while counterfactual signals capture *external evidence* that the video actually helps the prediction. Together, these provide a robust, interpretable measure of visual grounding we refer to as reliability.

### 3.1 FEATURE-BASED SIGNALS: INTERNAL RELIANCE

We compare the decoder's hidden states and attention patterns with and without video input.

- **Hidden state sensitivity.** If the decoder hidden state at step $t$ changes direction substantially when the encoder input is removed, the token is influenced by the video. We build on prior work (Li et al., 2017; 2024) by quantifying this directional change as the normalized angle between the hidden states with video ($h_t^{\text{vid}}$) and with masked input ($h_t^0$):

$$s_t^{\text{hid}} = \pi^{-1} \arccos\left( \frac{\langle h_t^{\text{vid}}, h_t^0 \rangle}{\|h_t^{\text{vid}}\| \, \|h_t^0\|} \right), \tag{1}$$

  where $\langle \cdot, \cdot \rangle$ denotes the inner product and $\|\cdot\|$ the Euclidean norm. A larger $s_t^{\text{hid}}$ indicates a stronger change in orientation, and thus stronger reliance on the video input. The division by $\pi$ normalizes the angular distance (in radians) to $[0, 1]$ for scale consistency.

- **Cross-attention usage.** Decoder cross-attention $A_{t \to i}^{(\ell,h)}$ at layer $\ell \in [1, L]$ and head $h \in [1, H]$ (from decoder token $t$ to encoder position $i$) is aggregated across all layers and heads ($L$: total decoder layers, $H$: attention heads). We compute the mean attention mass directed to video encoder positions and subtract the corresponding mass under a null (masked) encoder input, with per-sequence rescaling:

$$s_t^{\text{attn}} = \text{scale}\left( \frac{1}{LH} \sum_{\ell=1}^{L} \sum_{h=1}^{H} \sum_{i \in \text{video}} A_{t \to i}^{(\ell,h)} - \frac{1}{LH} \sum_{\ell=1}^{L} \sum_{h=1}^{H} \sum_{i \in \text{masked}} A_{t \to i}^{(\ell,h)} \right), \tag{2}$$

  where $\text{scale}(\cdot)$ applies quantile scaling to $[0, 1]$ across all decoder tokens $t$ in the same output sequence.
  A larger $s_t^{\text{attn}}$ indicates stronger reliance on grounded (video) information.

### 3.2 COUNTERFACTUAL SIGNALS: EXTERNAL EVIDENCE

While feature-based signals measure how much the decoder state *changes* when video is removed, they do not yet capture whether the video actively *helps* the chosen prediction. To complement the hidden state sensitivity measure, we introduce **counterfactual signals**, which quantify the *probability advantage* of real video compared to wrong or no video.

**Setup.** We run three parallel decoder passes, all conditioned on the same prefix $c_t$: (i) with the true video $x$, giving $p_{\text{vid}}(\cdot \mid c_t, x)$, (ii) with no video $\varnothing$ (cross-attention disabled), giving $p_0(\cdot \mid c_t)$, and (iii) with a mismatched video $x'$, giving $p_{\text{mis}}(\cdot \mid c_t, x')$.

Let $y_t$ be the token chosen under the clean pass with $x$. We then define the counterfactual[1] distribution as

$$p_{\text{cf}}(y_t \mid c_t) = \max\left\{ p_0(y_t \mid c_t), \, p_{\text{mis}}(y_t \mid c_t, x') \right\}. \tag{3}$$

This counterfactual approach is related to Chlon et al. (2025), but whereas they consider multiple counterfactuals, we focus on the single strongest alternative.

**Signal definitions.** From these passes we compute five complementary measures of video contribution (deltas and margins)[2]:

$s_t^{\text{log}} = \log p_{\text{vid}}(y_t) - \log p_{\text{cf}}(y_t)$      (log-probability margin);

$s_t^{\text{logit}} = \text{logit}\, p_{\text{vid}}(y_t) - \text{logit}\, p_{\text{cf}}(y_t)$      (scale-stable logit margin);

$s_t^{\text{prob}} = \sigma(s_t^{\text{log}})$      (normalized probability gain);

$\Delta_t^{\text{clean}} = p_{\text{vid}}(y_t) - p_0(y_t), \ \Delta_t^{\text{mis}} = p_{\text{vid}}(y_t) - p_{\text{mis}}(y_t)$      (advantages vs. no-video / mismatched);

$p_{\text{vid}}(y_t), \ p_0(y_t)$      (raw probabilities, optional).

---

[1] We take the maximum rather than the mean to avoid false positives: if either the no-video or the mismatched-video condition already explains $y_t$ nearly as well as the true video, then the token should be treated as ungrounded.

[2] Margins capture relative strength across scales, while deltas provide absolute probability gains; using both gives complementary views of video contribution.

**Interpretation.** All measures increase when the correct video makes the token more likely than either counterfactual.[3] The scores $s^{\log}$ and $s^{\mathrm{logit}}$ capture raw margins on log- and logit-scales, robust to small changes in scale. $s^{\mathrm{prob}}$ maps these into a $[0, 1]$ probability advantage, interpretable as a "confidence boost." $\Delta^{\mathrm{clean}}$ and $\Delta^{\mathrm{mis}}$ isolate the effect of removing vs. replacing the video, helping diagnose whether errors come from lack of signal or misleading signal.

**Role in reliability.** The counterfactual signal captures whether a token depends on visual input. Reliability is high when $p_{\mathrm{vid}}(y_t)$ exceeds the corresponding counterfactual probabilities, and low otherwise, that is, when the token is no more likely under the correct video than under its counterfactual variants. This directly links hallucination risk to the model's *under-use* of visual evidence.

## 3.3 Token-level and Sentence-level fusion of reliability signals

● **Token-level fusion:** For each token, we have two types of reliability signals:

1. *feature-based* signals $\mathbf{h}_t = \left[s_t^{\mathrm{hid}}, s_t^{\mathrm{attn}}\right]$, weighted by $\mathbf{w}_{\mathrm{fb}}$,

2. *counterfactual* signals $\mathbf{g}_t = \left[s_t^{\log}, s_t^{\mathrm{logit}}, s_t^{\mathrm{prob}}, \Delta_t^{\mathrm{clean}}, \Delta_t^{\mathrm{mis}}, p_{\mathrm{vid}}(y_t), p_0(y_t)\right]$, weighted by $\mathbf{w}_{\mathrm{cf}}$.

A **token-level reliability** score $r_t$ is then computed by a linear fusion with sigmoid activation:

$$r_t = \sigma\left(\mathbf{w}_{\mathrm{fb}}^\top \mathbf{h}_t \ + \ \mathbf{w}_{\mathrm{cf}}^\top \mathbf{g}_t \ + \ b\right). \tag{4}$$

● **Sentence-level fusion:** Since hallucination supervision is only available at the sentence level, we aggregate token reliabilities $\{r_t\}_{t=1}^T$ into a fixed-size feature vector using *early pooling*. For each sequence, we compute:

$$R_{\mathrm{tail\text{-}}q} = \frac{1}{\lceil qT \rceil} \sum_{t \in \mathrm{lowest}\ q\%} r_t \ (\text{average over lowest } q\% \text{ tokens})$$

where $qT$ denotes the number of tokens corresponding to the lowest $q\%$ of token reliabilities in the sequence (i.e., $\lceil qT \rceil$ tokens). We primarily rely on tail-10 pooling ($R_{\mathrm{tail\text{-}}10}$). Other options include $R_{\mathrm{mean}}$ (average over tokens), $R_{\mathrm{harm}}$ (harmonic mean), or $R_{\mathrm{min}}$ (minimum token reliability).

The resulting **sentence-level reliabilities** $R_{\mathrm{tail\text{-}}q}, \{R_{\mathrm{mean}}, \dots\}$ serve as inputs to downstream classifiers or regressors, enabling consistent comparison across utterances of varying length without masking.

## 3.4 Text only baselines and combination with grounding-based signals

To compare reliabilities across datasets and models, we fit a monotone mapping from the reliability deficit to the hallucination rate CHAIR:[4]

$$\mathrm{CHAIR} \approx \mathrm{ISO}\left(1 - R_*\right), \tag{5}$$

where $R_*$ is a chosen pooled statistic (e.g., tail-10%), and ISO denotes isotonic regression on a held-out split.

We benchmark reliability against text-only proxies computed from the clean run: **confidence** $C_t = p_{\mathrm{vid}}(y_t)$, **token entropy** $H_t = -\sum_w p_{\mathrm{vid}}(w) \log p_{\mathrm{vid}}(w)$, and **self-perplexity** $\mathrm{PPL}_t = 1/p_{\mathrm{vid}}(y_t)$ (perplexity under the model's own predictive distribution), pooled with the same pooling operator ($R_{\mathrm{tail\text{-}}10}$). Finally, we also evaluate a **combined META** variant that concatenates grounding-based signals with these text-side baselines (confidence, entropy, perplexity) and fits a linear layer on the pooled statistics. This allows us to assess whether our visual grounding signal provides complementary information beyond existing text-only measures.

---

[3]Here, a counterfactual means either removing the video input or replacing it with mismatched visual content.

[4]CHAIR (Caption Hallucination Assessment with Image Relevance) is a standard hallucination metric, here adapted to SLT by checking entity overlap between generated text and visual content (Rohrbach et al., 2018). Details can be found in Appendix F.2.

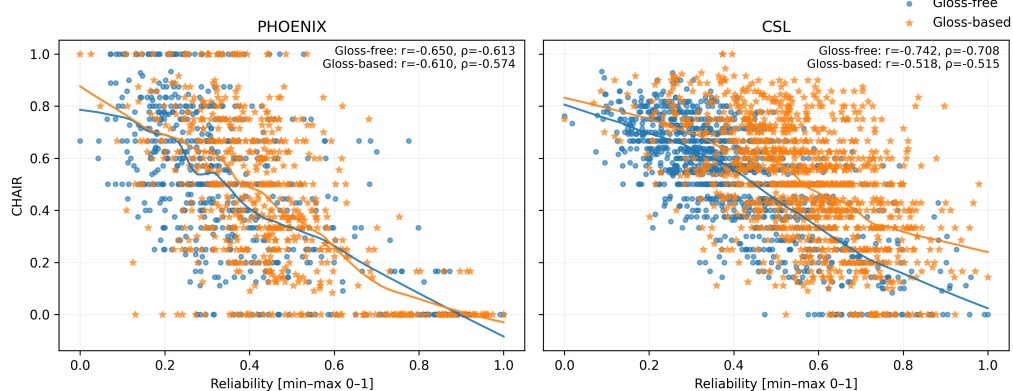

Figure 2: Correlation between reliability ($R_{\text{tail-10}}$) and hallucination rates (CHAIR) on PHOENIX-2014T (PHOENIX) and CSL-Daily (CSL). Higher reliability aligns with fewer hallucinations for both gloss-free and gloss-based models.

## 4 EXPERIMENTS AND RESULTS

### 4.1 EXPERIMENTAL DESIGN

Experiments are conducted on two datasets, PHOENIX-2014T (DGS→DE) (Camgoz et al., 2018) and CSL-Daily (CSL→ZH) (Zhou et al., 2021), using two competitive and reproducible systems: SpaMo, a gloss-free (**GF**) SLT model (Hwang et al., 2025), and TwoStream-SLT, a gloss-based (**GB**) model (Chen et al., 2022).

First, reliability features are extracted following the methodology described in Section 3, with the detailed algorithm provided in Appendix D.

Next, we train regression models for hallucination detection and for predicting CHAIR from these features, yielding reliability weights. The hallucination detection model is trained to identify hallucinated outputs, leveraging both our grounding-based reliability signals (visual features) and the baseline features described in Section 3.4, all integrated within the **META (ours)** framework. Accuracy is computed at a 0.5 threshold against the CHAIR-based detection label, where samples with $\text{CHAIR} > 0$ are labeled as 1 (hallucinated) and others as 0. Isotonic regression is employed to predict both CHAIR and its complement $(1 - \text{CHAIR})$. Evaluation metrics include AUROC (AUC), AUPRC (AP), and ACC@0.5 (ACC). The resulting weights are further examined for their transferability across datasets and model families.

Finally, reliability behavior is analyzed under both visual and linguistic degradations. Stress tests include Gaussian noise, temporal downsampling, and random frame dropping on the visual side, as well as wrong-start prompt prefixes to assess language-model dominance during decoding. In addition to CHAIR, the analysis tracks how BLEU (Papineni et al., 2002), an $n$-gram precision metric, and chrF (Popović, 2015), a character-$n$-gram F-score sensitive to morphological variation, evolve under these perturbations and how their trends correlate with reliability.

Evaluation includes correlations between reliability and CHAIR (Pearson, Spearman), regression fits ($R$ against $1 - \text{CHAIR}$), calibration robustness under cross-dataset/model transfer, and ablations comparing feature vs. counterfactual signals and pooling operators (mean vs. tail-10%). Implementation details appear in Appendix F.

### 4.2 RESULTS

**Low visual grounding as hallucination signal.** We begin by asking whether our reliability signal is meaningfully aligned with hallucination. Figure 2 shows a clear negative correlation between reliability ($R_{\text{tail-10}}$) and CHAIR across datasets and model types: as grounding usage (reliability) increases, hallucination decreases. This confirms that our measure captures an essential property of

|  | CSL GB | CSL GF | PHOENIX GB | PHOENIX GF |
|---|---|---|---|---|
| **Detection (AUC / AP / ACC)** | | | | |
| **Grounding (ours)** | **0.803 / 0.991 / 0.963** | 0.951 / 0.998 / **0.970** | **0.827 / 0.954 / 0.899** | 0.918 / **0.986 / 0.938** |
| **META (ours)** | **0.865 / 0.994 / 0.753** | **0.963 / 0.999 / 0.883** | **0.899 / 0.978 / 0.864** | **0.940 / 0.992 / 0.872** |
| Confidence | 0.647 / 0.976 / 0.508 | 0.953 / 0.998 / 0.868 | 0.618 / 0.887 / 0.645 | **0.926** / 0.989 / 0.860 |
| Perplexity | 0.609 / 0.980 / 0.610 | 0.952 / 0.998 / 0.842 | 0.595 / 0.892 / 0.561 | 0.888 / 0.979 / 0.713 |
| Token entropy | 0.729 / 0.985 / 0.665 | **0.958** / 0.998 / 0.866 | 0.639 / 0.900 / 0.581 | 0.871 / 0.975 / 0.731 |
| **Regression (Pearson / Spearman / ISO)** | | | | |
| **Grounding (ours)** | **-0.439 / -0.426 / 0.465** | -0.682 / -0.680 / 0.722 | **-0.458 / -0.400 / 0.493** | **-0.623 / -0.590 / 0.650** |
| **META (ours)** | **-0.518 / -0.515 / 0.543** | **-0.736 / -0.705 / 0.755** | **-0.610 / -0.574 / 0.637** | **-0.650 / -0.613 / 0.675** |
| Confidence | -0.253 / -0.269 / 0.294 | **-0.685** / -0.657 / 0.714 | -0.344 / -0.308 / 0.430 | -0.612 / -0.578 / 0.637 |
| Perplexity | -0.079 / -0.082 / 0.120 | -0.594 / -0.661 / 0.719 | -0.189 / -0.229 / 0.296 | -0.400 / -0.542 / 0.609 |
| Token entropy | -0.158 / -0.143 / 0.195 | -0.670 / **-0.698 / 0.748** | -0.275 / -0.273 / 0.342 | -0.468 / -0.544 / 0.625 |

Table 1: Summary of performance across datasets (CSL-Daily (CSL) and PHOENIX-2014T (PHOENIX)) and models (Gloss-based (GB) and Gloss-free (GF)). Top block reports **detection** metrics (AUC / AP / ACC), bottom block reports **regression** metrics (Pearson / Spearman / ISO). Best and second best values per metric are **bold-faced**.

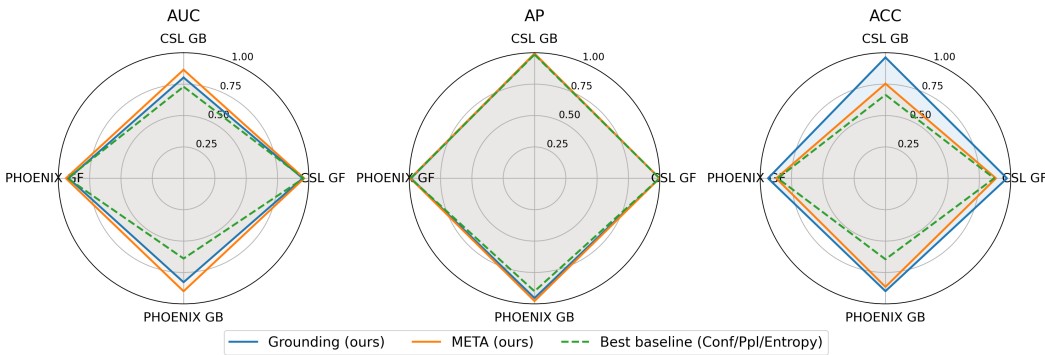

Figure 3: Detection metrics (AUC, AP, ACC) across datasets and models. We compare Grounding (ours), META (ours), and the best baseline (Confidence/Perplexity/Entropy) from Table 1.

hallucination detection. Interestingly, the correlation is consistently stronger in gloss-free models compared to gloss-based models. We attribute this gap to the higher hallucination rates in gloss-free systems: more frequent errors provide richer supervision for the reliability signal, making the negative slope sharper. In contrast, gloss-based systems hallucinate less often, yielding a weaker but still consistent trend.

These results establish the foundation for our subsequent analyses. Grounding usage is not just an auxiliary statistic, it already shows strong correlation with hallucination rates, especially in the more challenging gloss-free setting where visual grounding is the only safeguard against hallucinations. This validates its relevance as a hallucination signal and motivates the deeper evaluations that follow, including detection, regression, transfer, and stress tests.

**Hallucination detection (binary classification).** Table 1 upper block shows that our grounding-based reliability consistently improves over text-only baselines on hallucination detection. Across datasets and models, we gain up to +0.20 AUC and +0.30 ACC compared to entropy and perplexity, reaching near-perfect AP ($\approx 0.99$). We present the detailed gain over the other baselines in Table 2. The META (ours) variant, which fuses grounding with text-side uncertainty signals, achieves the strongest overall detection performance, demonstrating that grounding contributes complementary information beyond text probability-based baselines. At the same time, pure Grounding (ours) is slightly better than any text-only baseline in terms of accuracy, as highlighted in the Figure 3, showing that visual evidence can sharpen decision boundaries in ways text-only signals cannot. This advantage is especially critical in sign language translation, and even more so in the gloss-free case,

| Detection (A / P / C) | CSL GB | CSL GF | PHOENIX GB | PHOENIX GF |
|---|---|---|---|---|
| Confidence | A:+0.155 | A:-0.002 | A:+0.209 | A:-0.008 |
|  | P:+0.015 | P:-0.000 | P:+0.067 | P:-0.003 |
|  | C:+0.455 | C:+0.102 | C:+0.254 | C:+0.078 |
| Perplexity | A:+0.194 | A:-0.001 | A:+0.232 | A:+0.029 |
|  | P:+0.010 | P:-0.000 | P:+0.062 | P:+0.008 |
|  | C:+0.353 | C:+0.128 | C:+0.338 | C:+0.224 |
| Token entropy | A:+0.073 | A:-0.007 | A:+0.189 | A:+0.047 |
|  | P:+0.005 | P:-0.000 | P:+0.054 | P:+0.011 |
|  | C:+0.298 | C:+0.105 | C:+0.318 | C:+0.207 |

(a) Detection: $\Delta$(Grounding–baseline). A=AUC, P=AP, C=ACC.

| Regression (P / S / I) | CSL GB | CSL GF | PHOENIX GB | PHOENIX GF |
|---|---|---|---|---|
| Confidence | R:-0.186 | R:+0.003 | R:-0.114 | R:-0.011 |
|  | S:-0.157 | S:-0.022 | S:-0.092 | S:-0.011 |
|  | I:+0.171 | I:+0.007 | I:+0.063 | I:+0.013 |
| Perplexity | R:-0.360 | R:-0.088 | R:-0.269 | R:-0.223 |
|  | S:-0.344 | S:-0.019 | S:-0.171 | S:-0.048 |
|  | I:+0.346 | I:+0.003 | I:+0.197 | I:+0.041 |
| Token entropy | R:-0.281 | R:-0.012 | R:-0.183 | R:-0.155 |
|  | S:-0.283 | S:+0.018 | S:-0.127 | S:-0.046 |
|  | I:+0.270 | I:-0.026 | I:+0.151 | I:+0.025 |

(b) Regression: $\Delta$(Grounding–baseline). R=Pearson, S=Spearman, I=ISO.

Table 2: Summary of gains and drops over text-based signals. Columns correspond to dataset–model pairs (GB = Gloss-based, GF = Gloss-free). Blue indicates gain, orange indicates drop.

where no intermediate gloss supervision is available and reliable grounding in the continuous video stream is the only safeguard against hallucinations. More broadly, this underscores the need in multimodal generation to go beyond text-only uncertainty: hallucination detection benefits from explicitly checking whether outputs are supported by the input modality.

**Predicting CHAIR (regression).** On the regression task (see Table 1 bottom block), where CHAIR is approximated as a continuous target, text-based baselines achieve slightly higher correlations compared to using only **Grounding (ours)** (e.g., token entropy reaches $\rho = -0.698$ on CSL GF, perplexity $\rho = -0.542$ on PHOENIX GF), reflecting their strength as direct uncertainty estimators since entropy and perplexity are tied to token-level probabilities. Our grounding signal remains competitive, with Spearman correlations up to $\rho = -0.680$ (CSL GF) and isotonic fits of $0.72$–$0.65$ across datasets. Its counterfactual components exploit probability shifts between clean and perturbed video to provide an energy-based ranking that complements text baselines. The main observation is that while grounding alone does not fully match text-only uncertainty measures, it contributes complementary cues: text features capture uncertainty within the language model, while grounding verifies whether predictions are visually supported. Together, as shown by the **META (ours)** variant (reaching $\rho = -0.736$ on CSL GF and ISO up to $0.76$), this synergy yields the most reliable hallucination detection across datasets and transfer settings.

**Cross-dataset and cross-model generalization.** A linear calibration trained on one dataset transfers to another one with only a small drop in regression performance, and the same holds across gloss-free $\leftrightarrow$ gloss-based models. Some loss is expected, as the calibration reflects inevitable dataset- and model-specific biases. Nevertheless, **reliability** remains the strongest signal across transfers (the full transfer matrix is shown in the Appendix, Fig. 9). In Fig. 4(a) we present cross-dataset transfer within the same model family (GF vs GB). It remains close, as in CSL-GF $\rightarrow$ PHOENIX-GF, where the amount of hallucinations in both source and target might be similar and therefore the calibration signal learnable. As presented in Fig. 4(b), the transfer between gloss-based and gloss-free models leads to a more pronounced degradation in performance, indicating disparities in tokenization granularity and a reduction in exposure to linguistically informative samples (as gloss-free models tend to exhibit higher hallucination rates): gloss-based systems operate mostly at the word level to maintain gloss alignment, while gloss-free systems use subword units (Kudo & Richardson, 2018). Grounding signals align better with subunits, which clearly indicate which parts of words are guessed independently of video. Subunits level tokens often mark stems, subword boundaries which also correspond to visual units in sign language; glosses mostly represent content words, which are used as objects in CHAIR adaptation. This also suggests that transfer of the calibration to newer gloss-free architectures is more feasible, with smaller loss in terms of post-calibration performance, due to closer alignment with pretrained LLM subword tokenization. In Fig. 4(c), we also report the transfer gaps observed when transferring across both datasets and model types. The results confirm that the performance degradation is larger when reliability weights are transferred from gloss-based (GB) to gloss-free (GF) models, whereas the opposite direction (GF$\rightarrow$GB) exhibits smaller gaps.

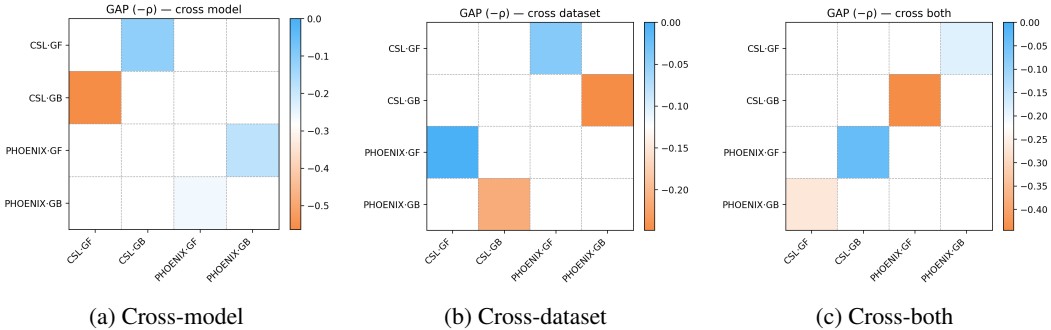

(a) Cross-model           (b) Cross-dataset          (c) Cross-both

Figure 4: Gaps in transfer reliability (negative Spearman correlation, $\rho$): (a) Cross-model within the same dataset, (b) Cross-dataset within the same model family, (c) Cross-both (dataset and model change). Blue = smaller drops, Orange = higher drops.

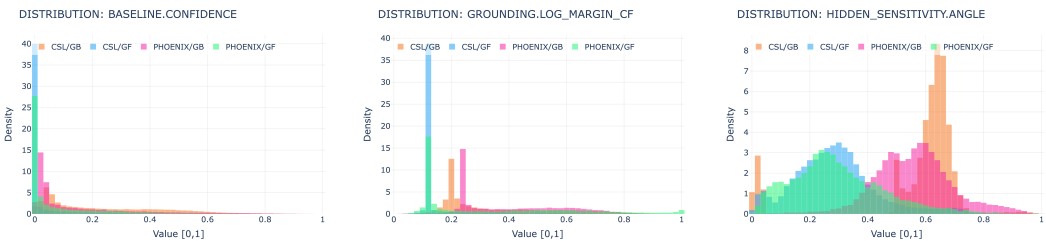

Figure 5: Token-level distributions (normalized [0,1]) for confidence ($-\log p$, left), grounding log-margin ($\log p(\text{video}) - \log p(\text{cf})$, middle), and hidden-state sensitivity angle (right).

**Why gloss-free systems hallucinate more often.** We compare token-level distributions of (i) *grounding log-margin* $\log p(\text{video}) - \log p(\text{cf})$, (ii) *hidden-state sensitivity angle*, and (iii) *confidence* ($-\log p$; near 0 means over-confident), as shown in Figure 5. **Log-margin:** GF is left-shifted on both datasets (most prominently on CSL), while GB maintains a longer high-margin tail, indicating that under counterfactual video, GF tokens remain plausible. **Hidden-state sensitivity angle:** GF concentrates at smaller angles ($\sim 0.20-0.35$ vs. $\sim 0.55-0.70$ for GB), indicating weaker state updates when the visual evidence changes. **Confidence-based hallucination signal** ($-\log p$)**:** GF mass sits near zero (over-confidence), whereas GB exhibits a heavier tail. The conjunction of *low margin + small angle + highly overconfident* occurs far more frequently in GF (subword tokenization), exactly where the model should lean on the video but does not; GB (word-level) yields more high-margin, video-responsive tokens. Consequently, text-only uncertainty reflects the LM's internal belief but misses cases where the *video fails to exert influence*. Our grounding signal evaluates this evidence, and the **META (ours)** fusion outperforms others by integrating uncertainty with explicit visual support. From this observation, we formalize the claim that *gloss-free* training increases hallucinations in SLT by increasing the prevalence of weak visual use (see proof in Appendix C).

**Reliability under visual degradation.** To test whether reliability reflects visual usage, we systematically degraded the input in two ways: (i) by adding random noise to video features, and (ii) by randomly dropping video frames. We then measured how the reliability measure (trained via regression on $1-\text{CHAIR}$) evolves under these perturbations. As shown in Figure 6, reliability consistently decreases as degradation intensity increases, tracking the rise in CHAIR and the decline in BLEU and chrF. The trend is not strictly monotonic, as raw regression outputs can fluctuate at higher degradation levels; this is expected since reliability is optimised to rank hallucination risk rather than to produce calibrated probabilities. Full degradation curves and numerical details are provided in Tables 3 and 4 in Appendix G.

**Prompt injection and language priors.** We further analyze model behavior under adversarially injected start tokens. A comprehensive set of representative samples spanning multiple injected prompts and both decoding settings is presented in Appendix H.2. These qualitative cases reveal a

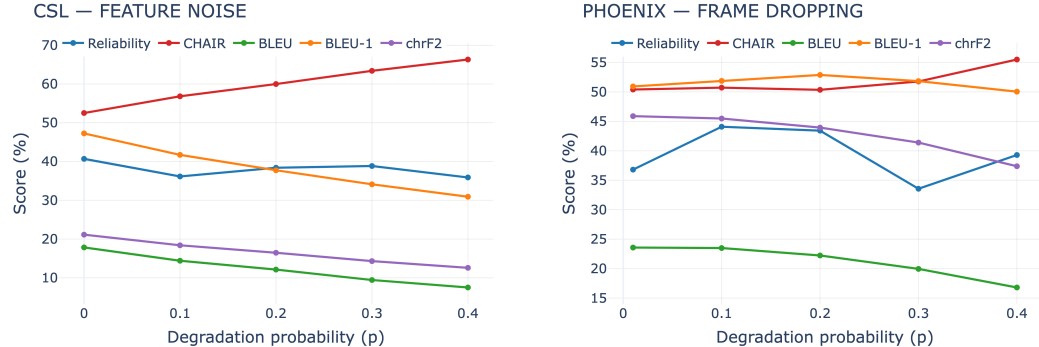

Figure 6: Reliability under visual degradation. Left: CSL with added feature noise. Right: PHOENIX with frame dropping. Reliability decreases with stronger degradation, with minor non-monotonic fluctuations, reflecting its role as a regression-based ranking signal rather than a calibrated probability.

consistent pattern: **gloss-free (GF)** models frequently continue the injected prefix and generate fluent continuations that follow linguistic priors, while **gloss-based (GB)** models trained with explicit visual supervision tend to ignore or rapidly override the injected cue and realign with the video evidence.

For instance, with the injected start "*ab den*" (*from the*), GF produces "*ab den nachmittagsstunden lässt der regen oder schnee an den alpen nach ...*" (*from the afternoon hours the rain or snow in the Alps eases ...*), faithfully extending the injected phrase but ignoring the signed content. In contrast, GB generates "*ab den ein kräftiges tief über der nordsee bringt uns ab morgen früh schneefälle ...*" (*a strong low over the North Sea brings us snowfall from tomorrow morning ...*), which discards the misleading prefix and aligns more closely with the reference.

This contrast substantiates our claim that *language priors dominate in gloss-free models*. In the absence of intermediate gloss supervision, GF relies heavily on text-only linguistic expectations at decoding, making injected prefixes act as strong attractors. In contrast, GB benefits from explicit supervision that reinforces cross-modal alignment, resulting in outputs that are more faithful to the signed input and less vulnerable to text-only biases.

## 5 CONCLUSION

Hallucination in multimodal generation cannot be fully explained by text-intrinsic signals such as confidence, entropy, or perplexity. While these signals capture uncertainty, they do not reveal whether outputs are actually grounded in external visual evidence. This limitation is especially critical in SLT, where the visual input is the true source language.

In this work, we propose a grounding usage based **reliability** measure as a new dimension for hallucination detection. Visual grounding usage quantifies how much each generated token is supported by visual input, and when combined with text-based signals (confidence, perplexity, or entropy), it improves hallucination detection by distinguishing visually grounded from visually unsupported tokens. Our experiments show that incorporating grounding usage with established text-based baselines improves predictive performance. Our analysis further reveals a systematic imbalance: gloss-free models trained without explicit visual supervision tend to under-use the visual input, which explains their higher hallucination rates relative to gloss-based models. The latter benefit from intermediate gloss supervision, which enforces stronger visual grounding. In other words, hallucinations in gloss-free SLT largely arise when the model "guesses" based on linguistic priors instead of truly attending to visual evidence.

In summary, a good multimodal model should strike the right balance between guessing (via language priors for fluency and grammaticality) and grounding (via visual evidence for factual correctness). Visual grounding alone is insufficient, but explicitly disentangling its contribution complements existing text-only signals and provides a more robust framework for hallucination detection.

## ACKNOWLEDGMENTS

This work was partially funded by the German ministry for education and research (BMBF) through project BIGEKO (grant number 16SV9093). KDC is supported by the Deutsche Forschungsgemeinschaft (DFG, German Research Foundation) – SFB 1102 Information Density and Linguistic Encoding. YG is supported by the German Federal Ministry of Research, Technology and Space (BMFTR) as part of the project TRAILS (01IW24005).

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

LIMITATIONS

Our study evaluates on two datasets (PHOENIX-2014T and CSL-Daily) using two architectures (SpaMo and TwoStream-SLT). While these cover both gloss-free and gloss-based paradigms across two sign languages, generalization to other sign languages, domains beyond weather forecasts, and newer architectures remains to be verified. The reliability framework requires three forward passes per sentence (clean, no-video, mismatched), tripling inference cost relative to standard decoding, which may limit applicability in real-time or resource-constrained settings. Although reliability can be applied without references at inference time, the fusion weights are trained using CHAIR labels derived from reference translations. Fully unsupervised calibration remains an open problem. Our adaptation of CHAIR to SLT operates at the content-word level and may not capture all hallucination types, such as temporal ordering errors, spatial confusions between similar signs, or hallucinated function words that alter sentence meaning. Human evaluation of token-level reliability scores would provide stronger validation but was not conducted in this work. Finally, cross-dataset and cross-model transfer, while competitive, shows non-negligible performance drops, particularly when transferring from gloss-based to gloss-free models. This suggests that reliability weights may need re-calibration when applied to new architectures or sign languages, which we leave for future work.

## A    LLM USAGE

We used LLMs solely for language polishing and minor phrasing edits.

## B    GROUNDING OR GUESSING? HYPOTHESES AND INTUITIONS

SLT has recently shifted from gloss-based pipelines (Camgoz et al., 2018; Chen et al., 2022) to gloss-free models (Hamidullah et al., 2022; Gong et al., 2024; Hamidullah et al., 2024; Hwang et al., 2025). While gloss annotations provide strong immediate supervision, they are costly to collect and limit scalability Camgoz et al. (2018); Zhou et al. (2021). Gloss-free approaches promise broader applicability, but they also introduce new challenges in grounding the translation in the visual signal (Guo et al., 2024; Rust et al., 2024).

Encoder–decoder models face a constant tension: tokens may be *grounded* in the source input or simply *guessed* from language priors. Function words and inflections can often be guessed predicted from context, but content words, numbers, names, entities require grounding in the encoder signal to avoid error.

**Why gloss-free SLT drifts.**   In gloss-based pipelines, explicit sign annotations provide strong visual anchors that keep the decoder aligned with the video. Gloss-free systems, however, pair video encoders directly with LLM decoders under weak sentence-level supervision. In the absence of dense visual anchors, the decoder defaults to and leans heavily on its language prior, producing fluent but unsupported tokens (i.e., hallucinations). The result is fluent text that is not grounded in the video, i.e., hallucination.

**Towards more hallucinations in SLT.**   Early SLT systems (Camgoz et al., 2018; 2020) mostly failed by omission: missing vocabulary or unseen expressions due to scarce training data. Even so, their outputs remained visually constrained by the gloss signal. Gloss-free models backed by strong LLMs, in contrast, exhibit a different failure mode: hallucinations dominate (Hwang et al., 2025). Entire entities absent from the video are introduced solely because the linguistic bias of the decoder fills in the gaps. So we currently observe a shift in the error profile: hallucinations dominate, where entities absent from the video are introduced by linguistic bias alone.

**Reliability as grounding measure.**   We hypothesize that hallucinations arise precisely when decoding relies on guessing rather than grounding. To test this, we introduce *reliability*, a token- and sentence-level measure of how much the decoder relies on the encoder signal of encoder usage. Figure 7 and Figure  8 show both mechanisms and the aggregate effect: Figure 7 at the token level, counterfactual signals highlight that content words (e.g., *freundlich*, *im Nord*) depend strongly on video, while function words remain largely language-driven; Figure 8 at the sentence level, lower

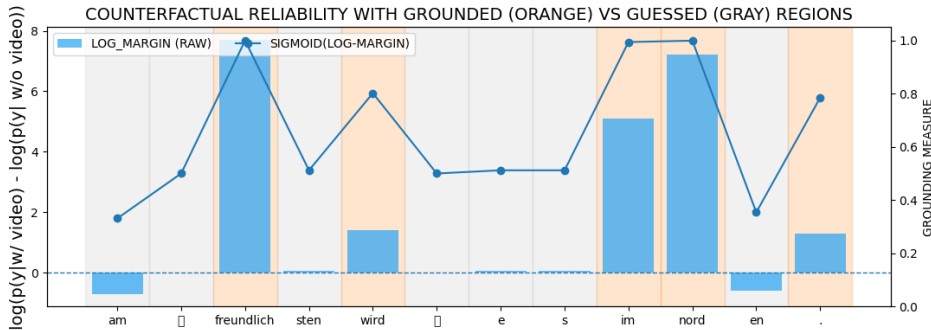

Figure 7: Token-level reliability. Content words show strong video dependence, whereas function words rely more on language priors.

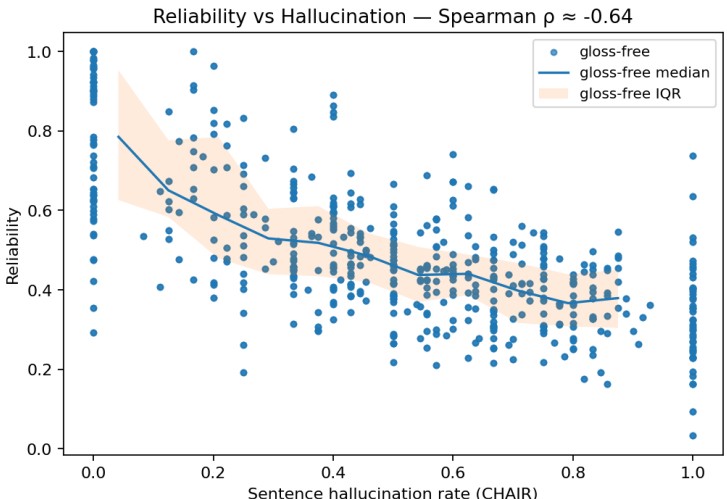

Figure 8: Sentence-level reliability vs. hallucination (CHAIR). Lower reliability correlates with higher hallucination.

reliability correlates with higher hallucination rates. Thus, reliability provides an explicit, quantifiable estimate of grounding confirming reliability as a direct estimate of grounding.

## C  SYMBOLIC PROOF: GLOSS-FREE SUPERVISION RAISES HALLUCINATIONS VIA WEAK VISUAL USE

We formalize the claim that *gloss-free* training increases hallucinations in SLT by increasing the prevalence of weak visual use.

**Random variables.** At a decoding step $t$, let $H \in \{0, 1\}$ indicate hallucination, $W \in \{0, 1\}$ indicate weak visual use (e.g., $r_t$ less than a threshold $\tau$), and $GF \in \{0, 1\}$ indicate gloss-free training.

**Assumptions.**

**(A1)** $\Pr(H{=}1) > 0$ (non-triviality)

**(A2)** $\Pr(H{=}1 \mid W{=}1) > \Pr(H{=}1 \mid W{=}0)$ (weakness raises hallucination)

**(A3)** $\Pr(W{=}1 \mid GF{=}1) > \Pr(W{=}1 \mid GF{=}0)$ (gloss-free raises weakness)

**(A4)** $H \perp\!\!\!\perp GF \mid W$ (mediation through $W$)

**Proposition 1** (Transitivity via a mediator)**.** *Under (A1)–(A4),*

$$\Pr(H{=}1 \mid GF{=}1) - \Pr(H{=}1 \mid GF{=}0) = \Big( \Pr(H{=}1 \mid W{=}1) - \Pr(H{=}1 \mid W{=}0)\Big)$$
$$\times \Big( \Pr(W{=}1 \mid GF{=}1) - \Pr(W{=}1 \mid GF{=}0)\Big) \quad (6)$$
$$> 0.$$

*Proof.* By the law of total probability and **(A4)**,

$$\Pr(H{=}1 \mid GF{=}g) = \sum_{w \in \{0,1\}} \Pr(H{=}1 \mid W{=}w) \Pr(W{=}w \mid GF{=}g).$$

Taking the difference between $g{=}1$ and $g{=}0$ and using $\Pr(W{=}0 \mid GF{=}g) = 1 - \Pr(W{=}1 \mid GF{=}g)$ yields

$$\Delta_H = \Big( \Pr(H{=}1 \mid W{=}1) - \Pr(H{=}1 \mid W{=}0)\Big) \Big( \Pr(W{=}1 \mid GF{=}1) - \Pr(W{=}1 \mid GF{=}0)\Big) \; > \; 0$$

$\square$

**Continuous variant.** Let $R \in [0,1]$ be *reliability*. If the mapping into probability $m(r) := \Pr(H{=}1 \mid R{=}r)$ is non-increasing, $R \mid GF{=}1$ is first-order stochastically dominated by $R \mid GF{=}0$, and $H \perp\!\!\!\perp GF \mid R$, then $\mathbb{E}[m(R) \mid GF{=}1] \geq \mathbb{E}[m(R) \mid GF{=}0]$ with strict inequality under strict dominance.

## D ALGORITHMS

We estimate reliability by comparing the model's predictions with and without visual input. Three passes are used: the true video (**video**), no visual encoder (**no-video**), and a shuffled or unrelated one (**mismatched-video**). From these, we derive **feature-based** signals (changes in hidden states and attention) and **counterfactual** signals (differences in token probabilities and logits). These are fused through a small logistic layer to obtain per-token reliabilities $r_t$. Early pooling (mean, tail-$q$, harmonic, min) yields sentence-level scores $\mathcal{R}$, indicating how much each translation relies on visual evidence.

## E EXTRA PLOTS

In Figure 9, we present the complete matrix illustrating the cross-transfer gap across datasets, models, and cross-both configurations. Overall, the performance drop remains relatively small; however, reliability weights trained gloss-based (GB) model outputs tend to generalize less effectively, likely due to reduced exposure to more informative samples and a higher tendency toward hallucinations.

Figure 10 summarizes the correlation performance of our visual signals relative to the strongest baseline for each metric. Similar to the detection results, our methods exhibit a stronger correlation with hallucinations when used as individual signals, and when combined with the text-based (uncertainty) baselines, they further enhance overall performance.

---

**Algorithm 1** RELIABILITYFROMVIDEO: full computation of sentence-level reliability

---

**Require:** SLT model $f_\theta$, video $x$, teacher-forced prefix tokens $c_{1:T}$, optional mismatched video $x'$
**Ensure:** Sentence-level pooled reliabilities $\mathcal{R} = \{R_{\text{mean}}, R_{\text{tail-}q}, R_{\text{harm}}, R_{\text{min}}\}$ and per-token $r_{1:T}$

   **Signals and parameters:**
   Feature-based signals at step $t$: $s_t^{\text{hid}}$, $s_t^{\text{attn}}$
   Counterfactual signals at step $t$: $s_t^{\text{log}}, s_t^{\text{logit}}, s_t^{\text{prob}}, \Delta_t^{\text{clean}}, \Delta_t^{\text{mis}}$
   Fusion weights: $\mathbf{w}_{\text{fb}} \in \mathbb{R}^2$, $\mathbf{w}_{\text{cf}} \in \mathbb{R}^5$, bias $b \in \mathbb{R}$ (default or pre-trained)
   Tail fraction $q \in (0,1)$ (e.g., $q = 0.1$), numerical constant $\varepsilon > 0$

---

   **Forward passes (teacher-forced):**
   $(z_{1:T}^{\text{vid}}, h_{1:T}^{\text{vid}}, A_{1:T}^{\text{vid}}) \leftarrow f_\theta(c_{1:T}, x; \texttt{use\_encoder} = \texttt{true})$
   $(z_{1:T}^{0}, h_{1:T}^{0}, A_{1:T}^{0}) \leftarrow f_\theta(c_{1:T}, \varnothing; \texttt{use\_encoder} = \texttt{false})$
   $(z_{1:T}^{\text{mis}}, \_, A_{1:T}^{\text{mis}}) \leftarrow f_\theta(c_{1:T}, x'; \texttt{use\_encoder} = \texttt{true})$       ▷ $x'$ is batch-shuffled or external
   $p_{\text{vid}} \leftarrow \text{softmax}(z^{\text{vid}})$, $p_0 \leftarrow \text{softmax}(z^0)$, $p_{\text{mis}} \leftarrow \text{softmax}(z^{\text{mis}})$

---

   **for** $t = 1$ **to** $T$ **do**
      **Feature-based (internal sensitivity):**
      $s_t^{\text{hid}} \leftarrow \pi^{-1}\arccos\big(\langle h_t^{\text{vid}}, h_t^0\rangle/(\|h_t^{\text{vid}}\|\|h_t^0\|)\big)$
      $\bar{A}_t^{\text{vid}} \leftarrow \frac{1}{LH}\sum_{\ell,h}\sum_i A_{t\to i}^{(\ell,h)}$,    $\bar{A}_t^0 \leftarrow \frac{1}{LH}\sum_{\ell,h}\sum_i A_{t\to i}^{0\,(\ell,h)}$
      $s_t^{\text{attn}} \leftarrow \texttt{QuantileScale}(\bar{A}_t^{\text{vid}}) - \texttt{QuantileScale}(\bar{A}_t^0)$
      **Counterfactual (external evidence):**
      $y_t \leftarrow \arg\max_w p_{\text{vid}}(w \mid t)$,    $p_{\text{cf}}(y_t) \leftarrow \max\{p_0(y_t), p_{\text{mis}}(y_t)\}$
      $s_t^{\text{log}} \leftarrow \log p_{\text{vid}}(y_t) - \log p_{\text{cf}}(y_t)$; $s_t^{\text{logit}} \leftarrow \text{logit}\, p_{\text{vid}}(y_t) - \text{logit}\, p_{\text{cf}}(y_t)$
      $s_t^{\text{prob}} \leftarrow \sigma(s_t^{\text{log}})$; $\Delta_t^{\text{clean}} \leftarrow p_{\text{vid}}(y_t) - p_0(y_t)$; $\Delta_t^{\text{mis}} \leftarrow p_{\text{vid}}(y_t) - p_{\text{mis}}(y_t)$
      **Token fusion:**
      $\mathbf{h}_t \leftarrow [s_t^{\text{hid}}, s_t^{\text{attn}}]^\top$; $\mathbf{g}_t \leftarrow [s_t^{\text{log}}, s_t^{\text{logit}}, s_t^{\text{prob}}, \Delta_t^{\text{clean}}, \Delta_t^{\text{mis}}]^\top$
      $r_t \leftarrow \sigma\big(\mathbf{w}_{\text{fb}}^\top \mathbf{h}_t + \mathbf{w}_{\text{cf}}^\top \mathbf{g}_t + b\big)$
   **end for**

---

   **Early pooling to sentence-level reliabilities:**
   $R_{\text{mean}} \leftarrow \frac{1}{T}\sum_{t=1}^T r_t$;    $R_{\text{tail-}q} \leftarrow \frac{1}{\lceil qT\rceil}\sum_{t\in \text{lowest } q\%} r_t$
   $R_{\text{harm}} \leftarrow \frac{T}{\sum_{t=1}^T \frac{1}{r_t+\varepsilon}}$;    $R_{\text{min}} \leftarrow \min_t r_t$
   **return** $\mathcal{R} = \{R_{\text{mean}}, R_{\text{tail-}q}, R_{\text{harm}}, R_{\text{min}}\}$ and $r_{1:T}$

---

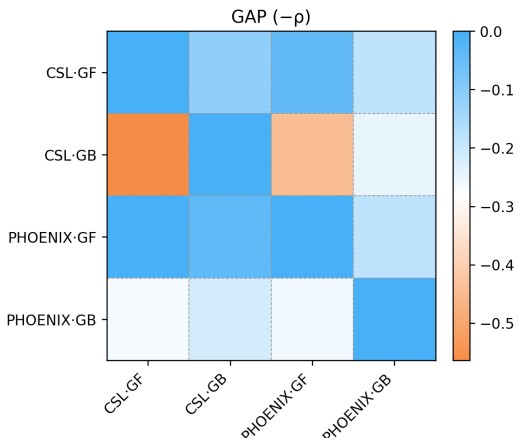

Figure 9: Full cross transfer gap.

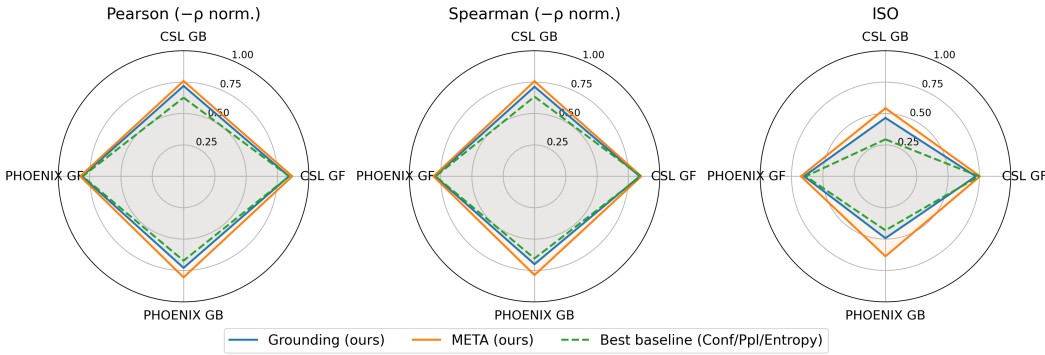

Figure 10: Radar visualization of correlation metrics (Pearson, Spearman, and Isotonic (ISO)) between CHAIR and reliability scores ($R_{\text{tail-10}}$), normalized to the $[0, 1]$ range.

## F  IMPLEMENTATION DETAILS

### F.1  GENERICS

We compute clean, no-video, and mismatched runs jointly in a single batched forward pass with shared key–value caches, with mismatches generated by in-batch shuffling. Probabilities are accumulated in FP32 with $\varepsilon = 10^{-12}$ for logarithms, while encoder–decoder blocks use AMP/FP16. Regression outputs are kept both raw and after sigmoid squashing. Cross-attention usage is normalized per sequence by mapping the 10/90% quantiles of mass to $[0, 1]$. Sentence scores are obtained by pooling token-level reliabilities with mean, harmonic mean, tail-10%, exponential moving average ($\alpha = 0.9$), or minimum. Calibration is applied per dataset/model/pool using either linear regression ($\hat{R} = \alpha R + \beta$) or isotonic regression against $1 - \text{CHAIR}$. Baselines (confidence, entropy, perplexity) are computed from the clean run; combined variants are also exported.

### F.2  CHAIR COMPUTATION AND LANGUAGE-SPECIFIC PREPROCESSING

To evaluate hallucinations we adopt the CHAIR metric (Rohrbach et al., 2018), which measures the proportion of hallucinated content words in the prediction relative to the reference. Since CHAIR relies on accurate identification of content tokens, we applied different preprocessing pipelines for **German** (PHOENIX-2014T data) and **Chinese** (CSL-Daily data) to account for linguistic differences.

**German (alphabetic languages).**   For German, we rely on `spaCy` for tokenization and part-of-speech (PoS) tagging, keeping only tokens whose PoS belongs to a restricted content set (*NOUN, PROPN, VERB, AUX, ADJ, ADV, NUM*). Tokens matching a stopword list (e.g., *und, der, ist*) are discarded. The remaining tokens are normalized by lemmatization and optionally stemmed using the `Snowball` stemmer (German/English), ensuring that inflected forms (e.g., *ginge, gegangen*) map to the same base form. This reduces sparsity and makes CHAIR counts more consistent.

**Chinese (logographic language).**   For Chinese, we cannot rely on whitespace or inflectional morphology. Instead, we: (i) normalize the script with `OpenCC` (Traditional → Simplified) when needed; (ii) normalize spacing around Chinese punctuation to avoid spurious tokens; (iii) tokenize using `jieba`; and (iv) apply a Chinese stopword list to remove frequent grammatical particles and function words. Since Chinese lacks inflectional morphology, no stemming is applied.

**CHAIR scoring.**   Given prediction and reference content tokens, we compute per-sentence hallucination as:

- **Instance-level CHAIR** ($\text{CHAIR}_I$): fraction of predicted content tokens that are absent or over-predicted compared to the reference.

The formulation from Rohrbach et al. (2018) is defined as:

$$\text{CHAIR}_i = \frac{\text{\# hallucinated object instances}}{\text{\# all mentioned object instances}}$$

where the object is here replaced by the content words.

This setup ensures that hallucination evaluation is *linguistically adapted*: German tokens are normalized via stemming/lemmatization, while Chinese tokens are segmented and filtered with script and stopword normalization. We use the instance-level CHAIR as label to train reliability linear weights.

# G ABLATIONS

## G.1 DEGRADATION UNDER VISUAL PERTURBATIONS

As presented earlier in Fig. 6, to assess whether reliability reflects visual usage, we systematically degraded the video input by adding Gaussian noise and randomly dropping frames. The reliability measure—trained via regression on $(1 - \text{CHAIR})$—was then evaluated under these perturbations. Reliability was rescaled to $[0, 1]$ for visualization, though correlations remain unaffected by this linear transformation.

Reliability consistently decreases as degradation increases, confirming its sensitivity to visual information, though slight non-monotonic fluctuations occur because it is optimized to *rank* hallucination risk rather than to yield calibrated probabilities. CHAIR and anti-BLEU-1 exhibit similar trends, with reliability following the same direction but with higher variance, consistent with its role as an anti-CHAIR ranking signal.

Overall results are presented in Tables 3 and 4.

| $p$ | Reliability | CHAIR | BLEU | BLEU-1 | chrF2 |
|------|-------------|-------|-------|--------|-------|
| 0.01 | 0.37 | 0.50 | 23.58 | 50.92 | 45.88 |
| 0.10 | 0.44 | 0.51 | 23.49 | 51.87 | 45.48 |
| 0.20 | 0.43 | 0.50 | 22.23 | 52.87 | 43.94 |
| 0.30 | 0.34 | 0.52 | 19.95 | 51.83 | 41.40 |
| 0.40 | 0.39 | 0.55 | 16.80 | 50.04 | 37.38 |

Table 3: Degradation on PHOENIX under frame dropping. Values are means over runs. Reliability and CHAIR are reported in raw scale; BLEU, BLEU-1, and chrF2 are on their standard scales.

| $p$ | Reliability | CHAIR | BLEU | BLEU-1 | chrF2 |
|------|-------------|-------|-------|--------|-------|
| 0.00 | 0.41 | 0.53 | 17.84 | 47.26 | 21.14 |
| 0.10 | 0.36 | 0.57 | 14.41 | 41.72 | 18.40 |
| 0.20 | 0.38 | 0.60 | 12.13 | 37.75 | 16.47 |
| 0.30 | 0.39 | 0.63 | 9.44 | 34.13 | 14.32 |
| 0.40 | 0.36 | 0.66 | 7.51 | 30.92 | 12.58 |

Table 4: Degradation on CSL under feature noise. Values are means over runs. Reliability and CHAIR are reported in raw scale; BLEU, BLEU-1, and chrF2 are on their standard scales.

## G.2 FEATURE IMPORTANCE AND POOLING

**Feature weights.** To better understand which reliability signals drive hallucination detection, we analyze the learned weights of our logistic regression head (Fig. 11). The model assigns the largest positive weight to the counterfactual *log_margin*, confirming that probability differences between clean and perturbed video inputs are the most informative cue. Smaller but consistent contributions are observed for *logit-margin*, *prob-margin*, and *attention-difference*, whereas *hidden-state sensitivity (hs_prob)* receives a small negative weight. This suggests that hidden-state changes capture some

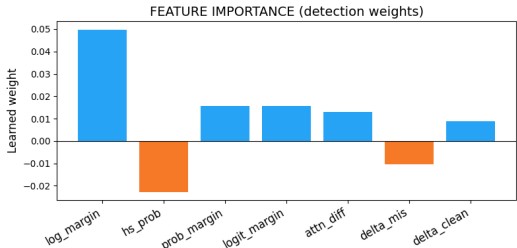

Figure 11: Learned feature weights for hallucination detection. Positive weights indicate stronger reliance of the classifier on the corresponding reliability signal.

grounding information, but they are less predictive than counterfactual margins. Delta-based features (*delta_clean*, *delta_mis*) contribute minimally once margin-based signals are included, indicating redundancy.

**Pooling strategies.** We further investigate the effect of *pooling* strategies across tokens (Table 5). We compare **mean pooling** ($R_{mean}$), and **tail10-pooling** ($R_{\text{tail-10}}$), which focuses on the least reliable 10% tokens.

Across datasets and models, $R_{\text{tail-10}}$ generally improves correlation with hallucination labels, especially for gloss-free (GF) models where weak grounding makes local unreliability highly informative. Gloss-based (GB) models also benefit, though the gap is smaller, indicating that tail pooling is particularly effective when hallucinations arise from localized failures of visual grounding.

| Dataset | Model | Metrics ($R_{mean} \mid R_{\text{tail-10}}$) | | |
|---|---|---|---|---|
| | | Pearson | Spearman | ISO |
| CSL | GB | **-0.439** \| -0.234 | **-0.426** \| -0.238 | **0.465** \| 0.282 |
| CSL | GF | -0.672 \| **-0.682** | **-0.678** \| -0.644 | **0.718** \| 0.704 |
| PHOENIX | GB | -0.361 \| **-0.458** | -0.368 \| **-0.340** | 0.453 \| **0.493** |
| PHOENIX | GF | -0.564 \| **-0.623** | -0.523 \| **-0.590** | 0.596 \| **0.650** |

Table 5: Comparison of pooling strategies. Each cell shows $R_{mean} \mid R_{\text{tail-10}}$. Tail pooling often yields stronger correlation with hallucinations, especially in gloss-free models.

## G.3 FEATURE INTERACTIONS WITH RAW PROBABILITIES

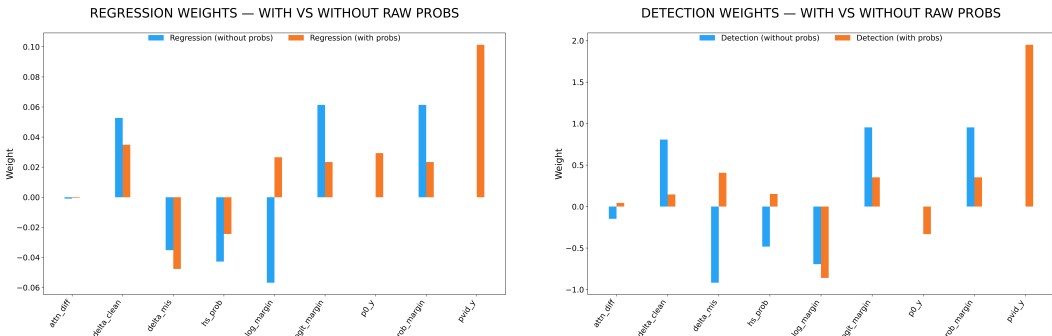

Figure 12: Regression (left) and detection (right) feature weights with vs. without raw probabilities.

As shown in Figure 12, adding raw probabilities $p_{\text{vid}}(y)$ yields a very strong but highly correlated predictor, which reduces the relative weight of complementary cues such as *hidden-state sensitivity*. This explains why grounding-based reliability remains competitive when used alone: it captures

visual reliance without being dominated by probability mass. In detection, hidden-state sensitivity contributes more strongly when considered in isolation without the raw probability, whereas in regression text-related signals dominate, highlighting complementarity rather than redundancy between probability-driven and state-driven evidence.

# H    QUALITATIVE ERROR ANALYSIS: GLOSS-FREE VS. GLOSS-BASED MODELS

We analyze a few overlapping references with outputs from both gloss-free (GF) and gloss-based (GB) models.

## H.1    GENERAL DISCUSSION

The qualitative analysis presented in Table 6 shows outputs that differ lexically or stylistically from the reference may still be semantically correct, while fluent but misleading sentences can conceal omissions, additions, or substitutions that critically alter meaning. Across cases, GF tends to produce translations that stay structurally close to the reference. Its errors are typically subtle, involving small lexical shifts or added modifiers, yet these can still distort key meteorological facts such as temperatures, locations, or weather conditions. GB, on the other hand, often generates more fluent and natural-sounding outputs. Omissions and hallucinated details are common, occasionally leading to forecasts that are polished in form yet misleading in content.

## H.2    QUALITATIVE: PROMPT INJECTION (EXTENDED)

This section provides an extended qualitative analysis of model behavior under adversarially injected start tokens. It includes a comprehensive set of representative samples covering multiple injected prompts and both decoding configurations. The examples complement the discussion in the main text, illustrating the systematic contrast between gloss-free (GF) presented in Table 7 and gloss-based (GB) models presented in Table 8: GF models often continue the injected prefix, producing linguistically fluent yet visually inconsistent continuations, whereas GB models, aided by explicit visual supervision, tend to suppress the injected bias and quickly realign with the visual evidence.

**Column descriptions:**

- Prompt: the injected adversarial prefix;
- Decoding: the generation strategy used (beam or greedy);
- Fooled: whether the model followed the injected prefix instead of the visual evidence;
- Prediction: the model's generated translation;
- Reference: the corresponding ground-truth sentence.

---

**Reference:** am tag breiten sich die teilweise kräftigen schneefälle weiter aus.

**GF:** zum teil breitet sich kräftiger schnee aus.
**GB:** dann kommen zum teil kräftige schneefälle auf.
**Analysis:** GF shortens and generalises: "kräftiger Schnee" instead of "kräftige Schneefälle" (Incorrect word), and drops "weiter aus" (Omission). GB reformulates with "dann kommen ... auf", preserving semantics. **Semantically correct**.

---

**Reference:** und es kann auch örtlich zum teil auch gefährlich glatt werden denn vor allen dingen in der mitte und auch im süden wird es frostig.
**GF:** sonst gibt es auch eher gefährliche straßenglätte vor allen dingen in der mitte und im süden wird es deutlich kühler werden.
**GB:** und es besteht erhöhte glättegefahr vor allen dingen in der mitte und auch im süden.
**Analysis:** GF paraphrases with "Straßenglätte" and adds "deutlich kühler werden" (Addition), losing the "örtlich" aspect (Omission). GB compresses but retains core meaning "erhöhte Glättegefahr" → **semantically adequate**.

---

**Reference:** der wind weht schwach bis mäßig aus südosten.

**GF:** der wind weht schwach bis mäßig aus nord bis nordost.
**GB:** der wind weht schwach bis mäßig.
**Analysis:** GF changes wind direction ("Südosten" → "Nord bis Nordost") = Incorrect word. GB omits wind direction entirely = Omission. Both distort crucial meteorological info.

---

**Reference:** ein ausläufer des tiefs über dem nordmeer lenkt allerdings von nordwesten wolkenfelder ins land.
**GF:** richtung westen machen sich tiefausläufer mit wolken bemerkbar.
**GB:** ein kleines tief über dem nordwesten lenkt wolken ins land.
**Analysis:** GF simplifies but misplaces direction ("Richtung Westen" vs. "von Nordwesten") = Incorrect word. GB alters "Ausläufer des Tiefs über dem Nordmeer" → "kleines Tief über dem Nordwesten" (Incorrect word), still partially correct but factual distortion.

---

**Reference:** dabei das wetter aber nicht unfreundlich höchstens mal im osten und auch im süden ein paar schauer.
**GF:** das milde und trockene wetter wird auch in der osthälfte und im süden ein bisschen schauer geben.
**GB:** am freundlichsten bleibt es noch im osten und süden hier und da schauer.
**Analysis:** GF exaggerates with "mild und trocken" (Addition), partially shifts tone. GB preserves intended contrast (friendly with occasional showers) → **semantically correct**.

---

**Reference:** am tag dreizehn grad bei dauerregen und einundzwanzig grad am oberrhein.

**GF:** am tag dreizehn grad an der ostsee und bis zu elf grad in oberfranken.
**GB:** am tag dreizehn grad im osten und einundzwanzig grad am oberrhein.
**Analysis:** GF fabricates new regions ("Ostsee", "Oberfranken") → Incorrect word + Addition. GB faithful to source with minor rephrasing → **semantically correct**.

---

**Reference:** der wind weht schwach bis mäßig an der nordsee und im bergland auch frischer wind.

**GF:** der wind weht schwach bis mäßig an der nordsee und auf den bergen teilweise auch frisch.
**GB:** der wind weht schwach bis mäßig an den küsten sowie aus süd bis südost.
**Analysis:** GF good paraphrase with minor lexical variation ("auf den Bergen" vs. "im Bergland") → **semantically correct**. GB introduces new direction "aus Süd bis Südost" Addition + Incorrect word.

---

**Reference:** auch im westen gibt es später wolkenlücken.

**GF:** im osten und westen mal dichtere wolken mal längere sonnenschein.
**GB:** das hoch das sich von westen bis in die mitte ausdehnt sorgt morgen verbreitet für sonnenschein.
**Analysis:** GF alters polarity: instead of "Wolkenlücken" it adds "mal dichtere Wolken" Incorrect word. GB hallucinates "Hochdruckgebiet" and causal explanation Addition, but preserves sunny outcome → partially correct.

---

Table 6: Comparative qualitative analysis of GF vs. GB translations. Errors are categorized using taxonomy of Vilar et al. (2006): Omission, Addition, and Incorrect word. Correct or near-equivalent paraphrases are marked as **semantically correct**.

| Prompt | Decoding | Fooled | Prediction | Reference |
|---|---|---|---|---|
| am tag | beam | Yes | am tag lässt der regen und schneefall an den alpen nach vor allem im osten und norden fällt noch etwas regen sonst ist es teilweise klar. | regen und schnee lassen an den alpen in der nacht nach im norden und nordosten fallen hier und da schauer sonst ist das klar. |
| *("in the morning / during the day")* | | | *(During the day, rain and snowfall over the Alps subside; mainly in the east and north a bit of rain still falls, otherwise it is partly clear.)* | *(Rain and snow subside over the Alps during the night; in the north and northeast, occasional showers fall, otherwise it is clear.)* |
| am tag | beam | Yes | am tag regnet es an den küsten etwas sonst mal sonne mal wolken vor allem in der mitte am freitag wechselhaftes wetter. | am donnerstag regen in der nordhälfte in der südhälfte mal sonne mal wolken ähnliches wetter dann auch am freitag. |
| *("in the morning / during the day")* | | | *(During the day, some rain on the coasts, otherwise sun and clouds alternating, especially in the center; on Friday, changeable weather.)* | *(On Thursday, rain in the northern half; in the south, sun and clouds alternating, similar weather also on Friday.)* |
| am tag | beam | Yes | am tag fließt von norden kühlere luft heran die für teilweise kräftige schneefälle und teilweise ergiebige regenfälle sorgt. | vom nordmeer zieht ein kräftiges tief heran und bringt uns ab den morgenstunden heftige schneefälle zum teil auch gefrierenden regen. |
| *("in the morning / during the day")* | | | *(During the day, cooler air flows in from the north, causing heavy snowfall and rain in places.)* | *(A strong low moves in from the North Sea, bringing heavy snowfall and some freezing rain from the morning hours.)* |
| am tag | greedy | Yes | am tag lässt der regen und schneefall an den alpen nach im osten und norden regnet es noch kräftig sonst ist es teilweise klar. | regen und schnee lassen an den alpen in der nacht nach im norden und nordosten fallen hier und da schauer sonst ist das klar. |
| *("in the morning / during the day")* | | | *(During the day, rain and snow over the Alps subside; in the east and north it still rains heavily, otherwise partly clear.)* | *(Rain and snow subside over the Alps during the night; in the north and northeast, occasional showers, otherwise clear.)* |
| am tag | greedy | Yes | am tag regnet es an den küsten etwas sonst mal sonne mal wolken und nur vereinzelt ein paar tropfen am freitag auch mal weniger. | am donnerstag regen in der nordhälfte in der südhälfte mal sonne mal wolken ähnliches wetter dann auch am freitag. |
| *("in the morning / during the day")* | | | *(During the day, some rain on the coasts, otherwise sun and clouds with only a few drops; on Friday, less.)* | *(On Thursday, rain in the northern half; in the south, sun and clouds alternating, similar weather also on Friday.)* |
| am tag | greedy | Yes | am tag überquert norddeutschland ein tief mit kräftigem schneefall und teilweise kräftigem regen. | vom nordmeer zieht ein kräftiges tief heran und bringt uns ab den morgenstunden heftige schneefälle zum teil auch gefrierenden regen. |
| *("in the morning / during the day")* | | | *(During the day, a low crosses northern Germany with heavy snowfall and rain in parts.)* | *(A strong low moves in from the North Sea, bringing heavy snowfall and some freezing rain from the morning hours.)* |
| ab den | beam | Yes | ab den nachmittagsstunden lässt der regen oder schnee an den alpen nach im osten und norden fällt noch etwas regen sonst ist es teilweise klar. | regen und schnee lassen an den alpen in der nacht nach im norden und nordosten fallen hier und da schauer sonst ist das klar. |
| *("from the ")* | | | *(From the afternoon hours, rain or snow over the Alps subside; in the east and north, some rain still falls, otherwise partly clear.)* | *(Rain and snow subside over the Alps during the night; in the north and northeast, occasional showers, otherwise clear.)* |
| ab den | beam | Yes | ab den donnerstags regnet es an den küsten mal sonne mal wolken vorsicht wegen glätte. | am donnerstag regen in der nordhälfte in der südhälfte mal sonne mal wolken ähnliches wetter dann auch am freitag. |
| *("from the ")* | | | *(From Thursday, rain on the coasts, alternating sun and clouds; caution due to slipperiness.)* | *(On Thursday, rain in the northern half; in the south, sun and clouds alternating, similar weather also on Friday.)* |
| ab den | beam | No | ab den nachmittagsstunden erreicht uns dann ein tief mit teilweise kräftigem schneefall und teilweise gefrierenden regen. | vom nordmeer zieht ein kräftiges tief heran und bringt uns ab den morgenstunden heftige schneefälle zum teil auch gefrierenden regen. |
| *("from the ")* | | | *(From the afternoon hours, a low reaches us bringing heavy snowfall and partly freezing rain.)* | *(A strong low moves in from the North Sea, bringing heavy snowfall and some freezing rain from the morning hours.)* |

Table 7: Nine representative prompt-injection cases for the GF (gloss-free) model with English translations (including prompt meaning) shown in italic below each example.

| Prompt | Decoding | Fooled | Prediction | Reference |
|---|---|---|---|---|
| am tag
*("in the morning / during the day")* | beam | No | am tag tiefer luftdruck bestimmt unser wetter .
*(During the day, low pressure determines our weather.)* | tiefer luftdruck bestimmt in den nächsten tagen unser wetter .
*(Low pressure will determine our weather in the coming days.)* |
| am tag

*("in the morning / during the day")* | beam | No | am tag dazu gibt es viele wolken die teilweise kräftige schauer und kräftige schauer vereinzelt auch gewitter bringen .
*(During the day, there are many clouds bringing partly heavy showers and isolated thunderstorms.)* | das bedeutet viele wolken und immer wieder zum teil kräftige schauer und gewitter .
*(That means many clouds and repeatedly partly heavy showers and thunderstorms.)* |
| am tag



*("in the morning / during the day")* | beam | No | am tag schwacher bis mäßiger an der see teilweise frischer wind aus süd bis südwest im nordwesten starke bis stürmische böen bei gewittern sturmböen .
*(During the day, weak to moderate, at the sea partly fresh wind from south to southwest; in the northwest, strong to stormy gusts and storm gusts in thunderstorms.)* | meist weht nur ein schwacher wind aus unterschiedlichen richtungen der bei schauern und gewittern stark böig sein kann .
*(Mostly only a weak wind blows from varying directions, which can become very gusty during showers and thunderstorms.)* |
| am tag
*("in the morning / during the day")* | greedy | No | am tag tiefer luftdruck bestimmt unser wetter .
*(During the day, low pressure determines our weather.)* | tiefer luftdruck bestimmt in den nächsten tagen unser wetter .
*(Low pressure will determine our weather in the coming days.)* |
| am tag

*("in the morning / during the day")* | greedy | No | am tag dazu gibt es viele wolken die teilweise kräftige schauer und kräftige schauer vereinzelt auch gewitter bringen .
*(During the day, there are many clouds bringing partly heavy showers and isolated thunderstorms.)* | das bedeutet viele wolken und immer wieder zum teil kräftige schauer und gewitter .
*(That means many clouds and repeatedly partly heavy showers and thunderstorms.)* |
| am tag


*("in the morning / during the day")* | greedy | No | am tag der wind weht schwach bis mäßig an der see auch frisch aus süd bis südwest im nordwesten bei gewittern sind stürmische böen möglich .
*(During the day, the wind blows weak to moderate, at the sea also fresh from south to southwest; in the northwest, stormy gusts possible in thunderstorms.)* | meist weht nur ein schwacher wind aus unterschiedlichen richtungen der bei schauern und gewittern stark böig sein kann .
*(Mostly only a weak wind blows from varying directions, which can become very gusty during showers and thunderstorms.)* |
| ab den

*("from the ")* | beam | Yes | ab den regen und schneefälle lassen die alpen im laufe des tages nach im nordosten regnet es mitunter noch kräftig sonst zeigen sich auch die sterne .
*(During the day, rain and snowfall over the Alps subside; in the northeast it still rains heavily at times, otherwise the stars are visible.)* | regen und schnee lassen an den alpen in der nacht nach im norden und nordosten fallen hier und da schauer sonst ist das klar .
*(Rain and snow subside over the Alps during the night; in the north and northeast, occasional showers, otherwise clear.)* |
| ab den


*("from the ")* | beam | Yes | ab den am donnerstag regnet es in der nordwesthälfte gebietsweise anfangs an den küsten mal sonne mal wolken und im wechsel gibt es dann am freitag ähnliches we
*(From Thursday, rain in the northwestern half, partly initially on the coasts alternating sun and clouds; similar weather continues on Friday.)* | am donnerstag regen in der nordhälfte in der südhälfte mal sonne mal wolken ähnliches wetter dann auch am freitag .

*(On Thursday, rain in the northern half; in the south, sun and clouds alternating, similar weather also on Friday.)* |
| ab den

*("from the ")* | beam | No | ab den ein kräftiges tief über der nordsee bringt uns ab morgen früh teilweise kräftige schneefälle und gefrierenden regen .
*(From a strong low over the North Sea brings us from early tomorrow partly heavy snowfall and freezing rain.)* | vom nordmeer zieht ein kräftiges tief heran und bringt uns ab den morgenstunden heftige schneefälle zum teil auch gefrierenden regen .
*(A strong low moves in from the North Sea, bringing heavy snowfall and some freezing rain from the morning hours.)* |

Table 8: Nine representative prompt-injection cases for the GB (gloss-based) model with English translations (including prompt meaning) shown in italic below each example.