# OpenReview forum: "Grounding or Guessing? Visual Signals for Detecting Hallucinations in Sign Language Translation"
_ICLR.cc/2026/Conference — ICLR 2026 Poster_

### Official Review · Reviewer_U3pX · 2025-10-24

**Soundness:** 3
**Presentation:** 3
**Contribution:** 3
**Rating:** 8
**Confidence:** 2

**Summary:**

This paper tackles the problem of hallucination in gloss-free sign language translation. SLT is a challenging task, which has been facilitated by the use of large language models (LLMs) and vision language models (VLMs) in recent years. However, these models tend to hallucinate. Gloss-level supervision can reduce hallucinations, by grounding the model with sign-level supervision, but glosses are expensive to annotate and label (note from reviewer: and introduce other problems). The authors of this paper propose a novel method to detect hallucinations and ground models in visual inputs, which reduces hallucinations and improves the results on gloss-free sign language translation.

**Strengths:**

- The paper is, for the most part, well-written and pleasant to read.
- The paper is well situated in related work.
- This is important work, as it improves the reliability of SLT models, especially gloss-free ones (which is critical because glosses can be considered harmful for SLT.)

**Weaknesses:**

- The paper contains acronyms that are not defined, e.g., AP.
- The methodology is rather math heavy and can be challenging to follow at times. It would benefit from some intuitive explanations of the equations.
- There is no statement on whether the signing community was consulted for this work. The authors should make a clear statement on their hearing/signing status.
- Appendix C and D are empty (probably because they contain only figures.) This layout problem should be addressed.

**Questions:**

- If we remove the subclause from the first line of your abstract, it reads "Hallucination is particularly critical in SLT." That is not the point you are trying to make, so please rephrase this.
- I recommend adding a statement about the dangers of using glosses for sign language translation at the end of the first paragraph of your introduction. They should not be used for sign language translation purposes: see [1]. This will further strengthen your paper since you are tackling gloss-free SLT.
- Does your system have merit in domains other than SLT to detect hallucinations?

---

> ### Author Response · Authors · 2025-11-26
> **Response to Reviewer U3pX**
>
> We thank the reviewer for their thoughtful comments. Addressing the other points made by the reviewer:
> ###### Weaknesses:
>
> ***The paper contains acronyms that are not defined, e.g., AP.***
>
> Thank you for pointing this out: In the revised version, all acronyms are defined.
> The methodology is rather math heavy and can be challenging to follow at times. It would benefit from some intuitive explanations of the equations.
>  We agree and have added explanations to all  the formulas.
>
> ***There is no statement on whether the signing community was consulted for this work. The authors should make a clear statement on their hearing/signing status.***
>
> Thank you for this comment. We have added an ethical consideration stating that our work uses only publicly available datasets, and that no new data collection or direct interaction with signers was conducted.
>
> ***Appendix C and D are empty (probably because they contain only figures.) This layout problem should be addressed.***
>
> Thank you for pointing this out:  Thank you for pointing this out. We have addressed this layout issue in the updated manuscript by adding descriptive details to ensure that each section in the appendix is complete and clearly presents its figures.
>
> ###### Questions:
>
> ***If we remove the subclause from the first line of your abstract, it reads "Hallucination is particularly critical in SLT." That is not the point you are trying to make, so please rephrase this.***
> We will revise  our abstract. Thank you for the suggestion.
>
> ***I recommend adding a statement about the dangers of using glosses for sign language translation at the end of the first paragraph of your introduction.
> They should not be used for sign language translation purposes: see [1]. This will further strengthen your paper since you are tackling gloss-free SLT.***
>
> We appreciate this suggestion. We will add a statement emphasizing that over-reliance on glosses can limit linguistic diversity and constrain translation quality, referencing [1] as suggested. This addition strengthens our motivation for focusing on gloss-free SLT.
>
> ***Does your system have merit in domains other than SLT to detect hallucinations?***
>
> Although we have not yet tested the method on other tasks, the proposed reliability measure is architecture- and modality-agnostic. Its underlying formulation can naturally extend to other vision–language tasks, where identifying content that is not visually supported is equally important. That said, SLT represents an extreme case: the entire source language is encoded in the visual signal, making visual grounding failures substantially more consequential than in typical multimodal tasks. As a result, while the method is broadly applicable, its value is particularly pronounced in SLT, where detecting hallucinations is both more challenging and more critical.

---

### Official Review · Reviewer_afii · 2025-10-31

**Soundness:** 3
**Presentation:** 2
**Contribution:** 2
**Rating:** 4
**Confidence:** 4

**Summary:**

This work investigates hallucination in sign language translation. The authors introduce a novel metric named reliability to measure the hallucination in SLT. The propose metric is based on two parts, i.e., internal model changes and output probability shifts. By adopting the CHAIR as the annotation on sentence-level, the proposed metric shows encouraging performance based on two SLT models on both benchmarks, i.e., PHOENIX-2014T and CSL-Daily.

**Strengths:**

It is the first work to investigate hallucination in sign language. The authors provide a novel metric to measure the reliability for SLT based on the visual input and text output. They also demonstrate the proposed reliability surpasses text-only baselines in detecting hallucinations.

**Weaknesses:**

1.The authors did not provide a clear analysis of the relationships about the three key concepts, i.e., sign language translation performance, the rate of hallucination and the proposed metric.

2.The key steps of the proposed metric is missing, such as how to calculate the weights. The appendix is also not completed, which is confusing. For specific issues, please refer to the descriptions in the Question section.

3.The authors claim that the hallucinations significantly impact the performance of SLT. However, there lacks data showing how hallucinations hurt translation quality. There is only limited comparative examples are given in the appendix, which is insufficient to support the main contributions.

4.Since the goal of mitigating hallucinations is to improve sign language translation performance, I think the authors should clarify the specific application directions of the proposed metric in current practical contexts.

5.The authors attribute the main reason of hallucinations to the poor performance of visual encoder. I think this point has been observed in the previous study [1]. It enhances visual representation capabilities by incorporating small pre-trained language models. Please provide more details about how to distinguish it from existing research in the manuscript.

[1] Zhigang Chen, Benjia Zhou, Jun Li, Jun Wan, Zhen Lei, Ning Jiang, Quan Lu, and Guoqing Zhao. 2024. Factorized Learning Assisted with Large Language Model for Gloss-free Sign Language Translation. In Proceedings of the 2024 Joint International Conference on Computational Linguistics, Language Resources and Evaluation (LREC-COLING 2024), pages 7071–7081, Torino, Italia. ELRA and ICCL.

**Questions:**

1.The authors provide the results on two SLT models. I suggest that the authors provide results on more open-source models. Additionally, the current method only considers cases where the input is video. Could it measures the different inputs, like poses?

2.All formulas in the manuscript lack recommended numbering.

3.There are notations in the equations lack descriptions.

In Line 154, $\pi^{-1}$ lack descriptions. Which part of the model does “hidden state” represent?

In Line 164, $scale$ lack descriptions. For $s_t^{attn}$, does a large result also indicate higher reliability? Are the set “video” and “masked” same?

4.In the 4 note, the authors provide a brief description about the calculation method of CHAIR. I think the authors should provide a more detailed description.

---

> ### Author Response · Authors · 2025-11-26
> **Response to Reviewer afii (part1)**
>
> We thank the reviewer for acknowledging the novelty of this work and for noting that our reliability measure effectively detects hallucinations beyond text-only baselines.
>
> ###### Weaknesses:
>
> ***1.The authors did not provide a clear analysis of the relationships about the three key concepts, i.e., sign language translation performance, the rate of hallucination and the proposed metric.***
>
> The relationship among sign language translation performance, hallucination rate, and our proposed reliability metric is illustrated in Figure 5 and discussed in §4.2. These results show that reliability correlates negatively with CHAIR (hallucination rate) and complements WER/BLEU (translation performance), highlighting its important contribution. While WER/BLEU capture linguistic correctness and CHAIR attempts to quantify visual ground errors by comparing textual object references generated by the model against ground truth labels, our reliability signal bridges the two by providing a reference-free measure of visual dependence. (Please also refer to Response (1.b) given to the reviewer bLfN). We will add a clarifying text in the Results section to make this critical connection explicit.
>
> ***2.The key steps of the proposed metric is missing, such as how to calculate the weights. The appendix is also not completed, which is confusing. For specific issues, please refer to the descriptions in the Question section.***
>
>
> Thank you for pointing this out. The revised manuscript now includes the previously missing definitions and provides a complete description of the reliability weighting process in section 4.2 Experimental design. All formulas are clearly numbered, and every variable is defined to ensure clarity and readability.
>
>
> ***3.The authors claim that the hallucinations significantly impact the performance of SLT. However, there lacks data showing how hallucinations hurt translation quality. There is only limited comparative examples are given in the appendix, which is insufficient to support the main contributions.***
>
>
> As shown in Figure 5, BLEU-1 is negatively correlated with CHAIR, confirming that higher hallucination rates coincide with lower translation performance. This demonstrates that hallucination directly reduces the model’s faithfulness to the visual input, even when textual metric remains high. Our counterfactual analysis (Figure 6) further separates cases where content tokens are predicted with versus without visual support, showing that errors in visually grounded content words are the primary source of hallucination and have a direct impact on translation quality. We will highlight this important relationship more explicitly in the revised manuscript.
>
>
> ***4.Since the goal of mitigating hallucinations is to improve sign language translation performance, I think the authors should clarify the specific application directions of the proposed metric in current practical contexts.***
>
>
> The point is well taken: The proposed reliability signal can serve as a reference-free confidence measure during inference. It can flag low-reliability outputs for human verification or guide adaptive decoding when visual grounding is weak. We will emphasize these practical applications in the Discussion section to clarify how the method can be integrated into real-world SLT systems.
>
>
> ***5.The authors attribute the main reason of hallucinations to the poor performance of visual encoder. I think this point has been observed in the previous study [1]. It enhances visual representation capabilities by incorporating small pre-trained language models. Please provide more details about how to distinguish it from existing research in the manuscript.***
>
> Chen et al. (2024) focuses on enhancing visual encoders through factorized learning assisted by language models. It is worth noting that the SpaMo model used in our work also performs visual block training, but it is not explicitly designed for hallucination reduction, even though it achieves improved results.
> In contrast, our work introduces a diagnostic reliability measure that quantifies the model’s reliance on visual input without altering its architecture. These two approaches are complementary: improved visual encoding can help reduce hallucinations, while our reliability metric provides a quantitative tool to measure and evaluate  the effect of such such improvements, i.e., it can directly assess how much a technique increases the model’s reliance on visual input.
>
> [1] Zhigang Chen, Benjia Zhou, Jun Li, Jun Wan, Zhen Lei, Ning Jiang, Quan Lu, and Guoqing Zhao. 2024. Factorized Learning Assisted with Large Language Model for Gloss-free Sign Language Translation. In Proceedings of the 2024 Joint International Conference on Computational Linguistics, Language Resources and Evaluation (LREC-COLING 2024), pages 7071–7081, Torino, Italia. ELRA and ICCL.

---

> > ### Author Response · Authors · 2025-11-26
> > **Response to Reviewer afii (part2)**
> >
> > ***Questions:***
> >
> > ***1.The authors provide the results on two SLT models. I suggest that the authors provide results on more open-source models. Additionally, the current method only considers cases where the input is video. Could it measures the different inputs, like poses?***
> >
> > We focused on the currently strongest SLT systems. The leading open-source models are not pose-based; even keypoint-based systems typically use a two-stream architecture where both video and keypoint sequences serve as visual features. Pose-based approaches do not achieve state-of-the-art performance in this setting. Our implementation follows this design, representing the best-performing and most widely adopted setup for SLT. Most other available models for the selected dataset are either non-pose-based, derived from the same architectures we use, or trained only on a single gloss-based dataset, making broader testing impractical.
> > ***2.All formulas in the manuscript lack recommended numbering.***
> > We have corrected this in the revised version.
> >
> > ***3.There are notations in the equations lack descriptions.***
> >
> > We provided more descriptions for the equations in the revised version.
> >
> > ***Which part of the model does “hidden state” represent?***
> >
> > Hidden state is just the output from the encoder/decoder layers.
> >
> > ***Missing descriptions comments***
> >
> > Thank you for the comment. All equations in the revised manuscript are now numbered, and all notations are clearly defined to ensure clarity and readability.
> > In this particular case we added: clarification of the \pi (normalisation), definition of all terms used in the signals formulas. All equations are now numbered, and all variables are defined.
> >
> > ***4.In the 4 note, the authors provide a brief description about the calculation method of CHAIR. I think the authors should provide a more detailed description.***
> >
> > We agree: We have expanded Appendix F to provide a comprehensive, step-by-step explanation of the CHAIR computation, including details on preprocessing and content-token selection for both languages.

---

### Official Review · Reviewer_bLfN · 2025-11-01

**Soundness:** 1
**Presentation:** 2
**Contribution:** 2
**Rating:** 2
**Confidence:** 5

**Summary:**

This manuscript aims to evaluate "hallucination" in sign language translation (SLT), defined as the extent to which the decoder generates token-level predictions without relying on visual information. To assess hallucination in SLT accurately, the manuscript proposes feature-wise signals, such as changes in feature orientation and intensity shifts in attention. Additionally, it leverages prediction discrepancies under noisy conditions to evaluate hallucination. Extensive experiments demonstrate that the proposed reliability indicator correlates with the "CHAIR" phenomenon in SLT, achieving impressive detection performance.

**Strengths:**

- S1. This manuscript is well-motivated, addressing the crucial issue of evaluating and improving the utilization of visual information in sign language translation (SLT), a topic that has garnered significant attention in recent years. The manuscript aims to design an indicator to shed light on this challenge.
- S2. The manuscript provides a comprehensive set of indicators for assessing the utilization of visual information, considering both the feature space and the output space.
- S3. The manuscript conducts thoughtful and thorough experiments on hallucination evaluation and detection.

**Weaknesses:**

- W1. The definition of hallucination in SLT is unclear. While hallucination in visual-language models (VLMs) is introduced due to the ambiguity in input range and output space, SLT has clearly defined output results and visual-language correspondences. The errors in hallucination can be directly measured using metrics like word error rate (WER) or similar set prediction metrics. Therefore, the introduction of the hallucination concept in SLT seems unnecessary.
- W2. As a translation task, many words are predicted based on contextual information. What we are truly concerned with are the words that can only be predicted by visual information. However, this manuscript does not address the distinction between different types of tokens in this context.
- W3. The insufficient use of visual information is a well-known issue in the sign language understanding community, and addressing this challenge is critical. While this manuscript attempts to detect hallucination, it does not propose any solutions for resolving this problem.
- W4. The practical applicability of the proposed method is limited. As shown in Figure 1, the proposed metric shows some correlation with "CHAIR" in SLT, which is used to demonstrate the effectiveness of the indicator. However, why not directly use "CHAIR" as the hallucination indicator? Furthermore, the high variation in the proposed metric suggests its limited practical value.
- W5. While Table 1 demonstrates that the proposed method achieves high hallucination detection accuracy, the process for obtaining ground-truth labels for detection remains unclear.

**Questions:**

- As mentioned in the weaknesses, I doubt the necessity of introducing the concept of hallucination in SLT, as it may mislead future research. However, I remain open to opinions and experimental evidence that could demonstrate the significance of hallucination in SLT.
- I also encourage the author to improve the writing and organization of this manuscript, and provide more valuable take home messages.

**Details Of Ethics Concerns:**

All experimental are conducted on public datasets.

---

> ### Author Response · Authors · 2025-11-26
> **Response to Reviewer bLfN (part1)**
>
> We thank the reviewer for their extensive feedback! We would further like to respond to some comments in more detail:
>
> ###### Weaknesses:
>
> ***Response (W1.a): The definition of hallucination in SLT is unclear.***
> We would like to clarify why the concept of hallucination is meaningful in SLT. In this work, we define hallucination in SLT as the generation of content tokens that are not supported by the signed video. This typically occurs when the decoder relies primarily on linguistic context or prior knowledge rather than on the visual information in the video. Since the video serves as the source language in SLT, every lexical unit in the translation should ideally be grounded in a corresponding visual cue. When a model produces words that appear plausible but cannot be traced back to the visual input, it is considered to be hallucinating. We have updated the introduction to include this precise definition.
>
> ***Response (W1.b): The errors in hallucination can be directly measured using metrics like word error rate (WER) or similar set prediction metrics***
>
> We respectfully disagree that standard metrics such as WER or set prediction metrics fully capture hallucination in SLT. These metrics measure translation errors relative to reference labels (and may incorrectly treat paraphrases as errors/hallucinations), but they do not evaluate whether the textual output is grounded in the visual input. In contrast our reliability measure is explicitly designed to capture this aspect in a reference-free manner by quantifying the extent to which the model depends on the signed video during inference. In this sense, it complements rather than competes with metrics like CHAIR or WER, similar to how uncertainty measures (e.g., entropy or confidence) are used to identify hallucinations in textual language models. We have updated the introduction to clarify this distinction clear and avoid any future confusion.
>
>
> ***Response (W1.c): Therefore, the introduction of the hallucination concept in SLT seems unnecessary / Question 1: I doubt the necessity of introducing the concept of hallucination in SLT.***
>
> We hope to have convinced the reviewer that our visual grounding-based  measure is well-motivated and advances SLT evaluation and hallucination detection beyond traditional text-only MT/ASR quality measures. We believe that studying hallucination in SLT is important, especially as large language models (LLMs) and multimodal foundation models are increasingly deployed in real-world scenarios such as live sign language interpretation for public services or virtual classrooms. These models can produce fluent outputs that appear correct but are not grounded in the visual input, largely due to the strong influence of the language model component, as reported in [1] and [9]. This mirrors challenges already observed in text-based LLMs. As SLT systems built on LLM architectures reach state-of-the-art performance, ensuring that their predictions faithfully reflect the visual content is critical for reliability, fairness, and user trust.
>
> ***W2. As a translation task, many words are predicted based on contextual information. What we are truly concerned with are the words that can only be predicted by visual information. However, this manuscript does not address the distinction between different types of tokens in this context.***
>
> Indeed, not all tokens in an SL translation depend equally on visual evidence. Function words and morphological markers are largely determined by linguistic context, whereas content words carry the core semantic meaning and should be grounded in the visual input. This distinction is explicitly captured in our novel definition of hallucination in SLT (see line 061-062 of the revised version). We define hallucination in SLT as the generation of content tokens that are not supported by the signed video. In computing CHAIR, we rigorously accounted for this by using only content words (see Appendix F.2) as object units in the hallucination rate calculation. As a result, the reported CHAIR values specifically reflect visually grounded content rather than context-driven language elements such as function words which can often be handled very well by a text-only based LLM. As shown in Figure 6, our counterfactual analysis clearly separates these cases. Hallucination occurs when content tokens that should rely on the visual modality are produced in text form but lack corresponding visual evidence. This also explains why our visual reliability signals consistently outperform text-only indicators such as entropy, confidence, or perplexity in detecting hallucination. Thus, by focusing on content tokens, we ensure that hallucination is measured meaningfully and that the evaluation accurately reflects the model’s reliance on visual input.

---

> > ### Author Response · Authors · 2025-11-26
> > **Response to Reviewer bLfN (part 2)**
> >
> > ***W3. The insufficient use of visual information is a well-known issue in the sign language understanding community, and addressing this challenge is critical. While this manuscript attempts to detect hallucination, it does not propose any solutions for resolving this problem.***
> >
> > We fully agree that the insufficient use of visual information is a long-standing and critical issue in the sign language understanding community. Our work directly addresses this challenge by focusing on detecting when a model fails to leverage visual cues, which we consider a necessary first step before developing mitigation strategies. The proposed reliability signals serve as a diagnostic measure, quantifying the degree of visual reliance or neglect during decoding. While this work does not by itself resolve the underlying cause of hallucination, it provides an essential foundation for future solutions. Long-term remedies will likely require complementary techniques such as multi-component mitigation frameworks, as explored in large language models [1,2,3]. Our approach is a prerequisite analytical step toward these broader interventions rather than a competing or incomplete fix. For example, reliability could be used as part of a scoring method in contrastive decoding. To the best of our knowledge, this is the first systematic study of hallucination in SLT.
> >
> > ***W4. The practical applicability of the proposed method is limited. As shown in Figure 1, the proposed metric shows some correlation with "CHAIR" in SLT, which is used to demonstrate the effectiveness of the indicator. However, why not directly use "CHAIR" as the hallucination indicator? Furthermore, the high variation in the proposed metric suggests its limited practical value.***
> >
> >
> > Please refer to Response W1.b for our clarification regarding the relationship between CHAIR and our reliability signal. We would like to emphasize that CHAIR and our proposed reliability signal serve fundamentally different purposes. CHAIR is a reference-based evaluation metric, requiring ground-truth annotations, and therefore cannot be used in real-time or unlabeled scenarios. In contrast, our reliability measure is reference-free and can be computed directly from model internals during inference. The correlation with CHAIR shown in Figure 1 serves as a validation step, demonstrating that our label-free signal aligns with a reference-based measure of hallucination. In essence, this shows that hallucinations detected by CHAIR are also captured by our measure, although CHAIR can incorrectly flag paraphrases as hallucinations and always requires references. Regarding variation, this is expected because reliability is a sensitivity-based diagnostic signal, not a deterministic score. Hallucination detection is inherently noisy due to automatic text processing, which introduces both false positives and false negatives. Achieving the reported correlation with a reference-based metric already demonstrates practical value. Furthermore, this variation provides a useful confidence range that can be thresholded for decision-making. We have updated the manuscript accordingly to clarify these points.
> > Reliability is sensitivity-based measure, meaning it captures a relative reaction or non-reaction to the visual content.
> > CHAIR: it measures how much of the predicted object (here, a lemmatized content word) is present in the reference. For CHAIR, if the model outputs something not present in the reference, this is counted as hallucination, without any direct access to the actual visual content or to the model’s internal representations.
> > Reliability: For each token, we determine whether it is sensitive to the visual content.
> > The ratio of grounded tokens to all tokens gives the reliability, while the ratio of non-grounded tokens to all tokens gives the hallucination rate.
> > Since we operate at the sentence level, we take into account that not all tokens need to be visually grounded. This explains why pooling the lowest-reliability tail yields a higher correlation with CHAIR: CHAIR already discards many tokens and focuses only on content words.
> > As we aim for a direct connection to the model’s internal mechanisms in order to support later mitigation strategies, having a tool like reliability (reference-free) derived from internal activations and functioning without labels at inference time is valuable. Introducing this notion of hallucination in sign language is therefore necessary, especially as recent gloss-free SLT systems increasingly rely on text-based generation, amplifying purely linguistic hallucinations in current SLT systems.

---

> ### Author Response · Authors · 2025-11-26
> **Response to Reviewer bLfN (part 3)**
>
> ***W5. While Table 1 demonstrates that the proposed method achieves high hallucination detection accuracy, the process for obtaining ground-truth labels for detection remains unclear.***
>
> Thank you for pointing this out. The ground-truth labels for hallucination detection were obtained using the CHAIR metric, following established practice from prior work [1]. Specifically, a sample is labeled as hallucinated if its CHAIR score is greater than zero, and non-hallucinated otherwise. For regression analysis, the continuous CHAIR score is used directly as the target. This procedure requires no manual annotation and provides a fully reproducible, label-based reference for evaluating our reference-free reliability signal. We have updated the Experimental Design section of the revised manuscript to explicitly and unambiguously describe this labeling process.
> ###### Questions:
> ***As mentioned in the weaknesses, I doubt the necessity of introducing the concept of hallucination in SLT, as it may mislead future research. However, I remain open to opinions and experimental evidence that could demonstrate the significance of hallucination in SLT.***
>
> Please refer to our responses to W1.b and  W1.c.
> For experimental evidence supporting the significance of hallucination in SLT, one can refer to recent papers such as [4],[5] and [8]. The reason these are recent is that current SLT research increasingly uses gloss-free approaches to leverage larger available datasets. These new models rely heavily on strong language components, which tend to dominate the weaker visual component, a long-standing issue in SLT. With the adoption of even stronger language models, hallucinations become further accentuated. Thus, we believe that this study is timely. Additional cross-confirmation can be found in studies such as [6] and the initial observations reported in [7]. However, none of the mentioned works conducted a systematic study of hallucination.
> In this work, we first formalize hallucination in SLT (see the updated definition). We then study its characteristics using internal model signals,showing that hallucination corresponds to identifiable patterns that can be detected without external references. We also provide an explanation for why gloss-free models are more prone to hallucinations.
> We hope to have addressed this concern convincingly.
>
> ***I also encourage the author to improve the writing and organization of this manuscript, and provide more valuable take home messages.***
>
> We have carefully revised the manuscript to improve both the writing and overall organization. The updated version incorporates the reviewer’s recommendations and also includes an improved and better-organized appendix.

---

> > ### Author Response · Authors · 2025-11-26
> > **Response to Reviewer bLfN (part 4)**
> >
> > References:
> >
> > [1] Yang, T., Li, Z., Cao, J., & Xu, C. (2025). Understanding and Mitigating Hallucination in Large Vision-Language Models via Modular Attribution and Intervention. In International Conference on Representation Learning (pp. 51546–51568).
> >
> > [2] Wang, X., Pan, J., Ding, L., & Biemann, C. (2024). Mitigating Hallucinations in Large Vision-Language Models with Instruction Contrastive Decoding. In Findings of the Association for Computational Linguistics: ACL 2024 (pp. 15840–15853). Association for Computational Linguistics.
> >
> > [3] Jiang, N., Kachinthaya, A., Petryk, S., & Gandelsman, Y. (2025). Interpreting and Editing Vision-Language Representations to Mitigate Hallucinations. In The Thirteenth International Conference on Learning Representations.
> >
> > [4] Artiaga, K., Kamila, S., Afli, H., Lynch, C., & Hasanuzzaman, M. (2025). Rethinking Sign Language Translation: The Impact of Signer Dependence on Model Evaluation. In Findings of the Association for Computational Linguistics: EMNLP 2025 (pp. 18379–18391). Association for Computational Linguistics.
> >
> > [5] Hamidullah, Y., Yazdani, S., Oguz, C., van Genabith, J., & España-Bonet, C. (2025, November). SONAR-SLT: Multilingual Sign Language Translation via Language-Agnostic Sentence Embedding Supervision. In Proceedings of the Tenth Conference on Machine Translation (pp. 301-313).
> >
> > [6] Chen, Z., Zhou, B., Li, J., Wan, J., Lei, Z., Jiang, N., Lu, Q., & Zhao, G. (2024). Factorized Learning Assisted with Large Language Model for Gloss-free Sign Language Translation. In Proceedings of the 2024 Joint International Conference on Computational Linguistics, Language Resources and Evaluation (LREC-COLING 2024) (pp. 7071–7081). ELRA and ICCL.
> >
> > [7] Zhang, B., Müller, M., & Sennrich, R. SLTUNET: A Simple Unified Model for Sign Language Translation. In The Eleventh International Conference on Learning Representations.
> >
> > [8] Shakib Yazdani, Yasser Hamidullah, Cristina España-Bonet, Eleftherios Avramidis, & Josef van Genabith. (2025). A Critical Study of Automatic Evaluation in Sign Language Translation.
> >
> > [9] Jian Ma, Wenguan Wang, Yi Yang, Weili Guan, & Feng Zheng. (2025). Hybrid model collaboration for sign language translation with VQ-VAE and RAG enhanced LLMs. (ICLR2025 Archive)

---

### Official Review · Reviewer_zRXC · 2025-11-03

**Soundness:** 3
**Presentation:** 3
**Contribution:** 3
**Rating:** 8
**Confidence:** 4

**Summary:**

The paper introduces a token-level reliability metric for hallucination detection in sign language translation (SLT), combining encoder sensitivity to input perturbations and internal state changes under counterfactual video masking. The metric is evaluated on PHOENIX-2014T and CSL-Daily using both gloss-based and gloss-free models.

**Strengths:**

-  Precisely identifies hallucination in SLT—especially in gloss-free settings—as a critical, under-addressed issue tied to visual grounding weakness.

- The token-level reliability score is architecture-independent, requiring no reference translations, making it broadly applicable across existing and future SLT systems.

- Combines encoder sensitivity (input robustness) and counterfactual masking (internal consistency) into a well-justified, interpretable proxy for visual grounding.

- Establishes a reusable evaluation tool that can become a standard metric in SLT research and development.

**Weaknesses:**

- Lacks testing on larger, more diverse, or continuous SLT benchmarks (e.g., How2Sign, OpenASL).
- Tested on a small set of architectures (primarily transformer-based).
- Unclear whether the proposed reliability score outperforms or complements prior art.

**Questions:**

Performance Impact on Gloss-Free SLT:
- The paper positions reliability as a diagnostic tool, yet provides no evidence that it leads to tangible improvements in gloss-free sign language translation. Can the authors demonstrate substantial gains (e.g., ≥1.0 BLEU)?

Cross-Architecture Validation:
- Claims of model-agnosticism are unsubstantiated. Can the authors apply the reliability score to ≥3 diverse gloss-free architectures (e.g., SignBT, STMC, BLT, or non-autoregressive variants) and show:  Consistent hallucination prediction (correlation, AUC)?

---

> ### Author Response · Authors · 2025-11-26
> **Response to Reviewer zRXC**
>
> We thank the reviewer for their thoughtful comments. Addressing the other points made by the reviewer:
>
> ###### Weaknesses:
> ***Lacks testing on larger, more diverse, or continuous SLT benchmarks (e.g., How2Sign, OpenASL).***
>
> We would like to test on larger and more diverse data sets. However, suitable broader benchmarks for our work are scarce and, to the best of our knowledge, currently unavailable. This is because (i) our research requires datasets with readily available gloss annotations, and (ii) these datasets must also be used to train existing SLT models. At present, only PHOENIX and CSL-Daily meet these criteria, which allows for direct comparison between gloss-based and gloss-free settings, a central focus of our paper. Unfortunately, we are not aware of any other datasets that satisfy these requirements (for example, in the case of How2Sign, gloss annotations are not yet available). That said, we appreciate your suggestion and hope to incorporate additional continuous SLT benchmarks in future work as soon as suitable datasets become available.
>
> ***Tested on a small set of architectures (primarily transformer-based).***
>
> We carefully chose transformer-based models as they are currently the strongest models used in gloss-free SLT  and serve as the dominant backbone in state-of-the-art systems. Our experiments therefore focus on this representative family, covering two distinct encoder–decoder configurations to assess stability across variants. As our approach is sensitivity-based and uses internal activations rather than architecture-specific components, it can be applied to other SLT models as long as they provide hidden states before producing output probabilities, which is the only requirement of our method.
>
> ***Unclear whether the proposed reliability score outperforms or complements prior art.***
>
> In §4.2 (Results), we show that the proposed reliability signals outperform prior text-only baselines such as entropy, confidence, and perplexity in detecting hallucination. We also show that the combined META method performs even better (line 361-363). This demonstrates that our approach complements and extends existing uncertainty-based methods (text-only).
>
> ###### Questions:
> ***The paper positions reliability as a diagnostic tool, yet provides no evidence that it leads to tangible improvements in gloss-free sign language translation. Can the authors demonstrate substantial gains (e.g., ≥1.0 BLEU)?***
>
> We emphasize that our grounding-based reliability measure is intended as a diagnostic tool rather than an optimization objective. Its major motivation is to assess the extent to which the model relies on the visual modality during decoding, rather than to directly improve BLEU. That said, we observed that samples with higher reliability scores consistently correspond to better translation accuracy and lower CHAIR, demonstrating that stronger visual grounding not only aligns with our diagnostic measure but also serves as an indirect signal of translation quality.
>
> ***Claims of model-agnosticism are unsubstantiated. Can the authors apply the reliability score to ≥3 diverse gloss-free architectures (e.g., SignBT, STMC, BLT, or non-autoregressive variants) and show: Consistent hallucination prediction (correlation, AUC)?***
>
> We appreciate the reviewer’s comment regarding cross-architecture validation. In this study, we focus on state-of-the-art encoder–decoder architectures, which are currently the dominant and highest-performing in SLT. We evaluate the proposed reliability signal on two widely used and competitive models within this scope. The consistent behavior of the reliability signal across these architectures, along with its evaluation in cross-model and cross-dataset settings (see Fig. 3c), demonstrates its effectiveness for the most relevant and performant models. While we do not claim full model-agnosticism, we have covered here two very distinct models and they were also selected on their reproducibility. We have added a brief clarification on the choice of models used here (see line 262 of the revised version) “...using two widely competitive and reproducible systems.”.

---

### Meta-Review · Area_Chair_Q9fz · 2025-12-09

**Summary:**

This paper proposes a token-level reliability measure that captures how much SLT models rely on visual input and shows that it correlates with hallucination and complements text-only uncertainty signals. Two reviewers were positive from the start, while others questioned the necessity of defining hallucination in SLT, the relation to CHAIR, and the clarity of key technical steps. The rebuttal clarified the definition, explained how labels are derived, completed missing formulas, and better distinguished reliability from reference-based metrics.

**Reviewer Concerns:**

The rebuttal effectively addressed concerns about the definition of hallucination in SLT, the distinction between reliability and CHAIR, the source of ground-truth labels, and the missing technical details in the formulas and appendix. It also clarified how reliability relates to translation quality and why gloss-free settings make this problem more pronounced. Remaining concerns include the limited architectural diversity in experiments, the lack of direct evidence that reliability can improve SLT performance when used in practice, and questions about how the method would operate in broader real-world settings.

**Reviewer Scores:**

Reviewer zRXC began with a score of 8 and expressed confidence in the contribution, so their score would likely remain at 8 or rise slightly after discussion. Reviewer bLfN started at 2 and, although the rebuttal clarified several points, they maintained strong reservations about the framing of hallucination in SLT, so their score might move only modestly to 3 or 4. Reviewer afii began at 4 and noted that many of their concerns were technical or expository; given the strengthened explanations in the rebuttal, their score could reasonably increase to around 5. Reviewer U3pX started at 8 and showed consistent enthusiasm, so their score would likely stay at 8.

---

### Decision · Program_Chairs · 2026-01-26

Accept (Poster)